

**Theoretical analysis of mixing in liquid clouds. Part IV: DSD evolution**

**and mixing diagrams**

Mark Pinsky, and Alexander Khain

Department of Atmospheric Sciences, The Hebrew University of Jerusalem, Israel

Submitted to

Atmospheric Chemistry and Physics

May 2017

Revised September 2017

Second revision: November 2017

Communicating author: Alexander Khain, The Hebrew University of Jerusalem, khain@vms.huji.ac.il

**Abstract**
Evolution of droplet size distribution (DSD) due to mixing between cloudy and dry
volumes is investigated for different values of the cloud fraction and for different initial DSD
shapes. The analysis is performed using a diffusion-evaporation model which describes time-
dependent processes of turbulent diffusion and droplet evaporation within a mixing volume.
Time evolution of the DSD characteristics such as droplet concentration, LWC and mean
volume radii is analyzed. The mixing diagrams are plotted for the final mixing stages. It is
shown that the difference between the mixing diagrams for homogeneous and inhomogeneous
mixing is insignificant and decreases with an increase in the DSD width. The dependencies of
normalized cube of the mean volume radius on the cloud fraction were compared with those
on normalized droplet concentration and found to be quite different. In case the normalized
droplet concentration is used, mixing diagrams do not show any significant dependence on
relative humidity in the dry volume.
The main conclusion of the study is that traditional mixing diagrams cannot serve as a
reliable tool for analysis of mixing type.
**Keywords:** turbulent mixing, droplet evaporation, DSD evolution, mixing diagram
**1.    Introduction**

The effects of mixing of cloudy air with surrounding dry air on cloud microphysics are still

the focus of many studies (see overview by Devenish et al., 2012). Processes of mixing are
investigated in observations (Yum et al., 2015; Bera at al., 2016a,b), Large Eddy Simulations
(Andrejczuk et al., 2009; Khain et al., 2017) and Direct Numerical Simulations (Kumar et al.,
2014, 2017). Processes of mixing and their effects on droplet size distributions were recently
investigated in a set of theoretical studies (Yang et al., 2016; Korolev et al., 2016 (hereafter,
Pt1); Pinsky et al., 2016 a,b).

The Pt1 presented analysis of conventional (classical) concept of mixing and introduced the

main parameters characterizing homogeneous and extremely inhomogeneous mixing. In the
classical concept two volumes, cloudy and droplet free one, mix within an unmovable adiabatic
mixing volume. At a monodisperse initial droplet size distribution (DSD), homogeneous mixing
leads to a decrease in droplet size and droplet mass content, while the number of droplets
remains unchanged. Extremely inhomogeneous mixing is characterized by decreasing the
number of droplets due to full evaporation of some fraction of droplets penetrating the initially
dry air volume while the DSD shape in the cloud volume remains unchanged. As a result of
extremely inhomogeneous mixing, droplet number decreases while the mean volume radii
remain unchanged. At a polydisperse DSDs, the extreme homogeneous mixing is characterized
by proportional changes in DSD for all droplet radii (Pt1). Since widely used mixing diagrams
describe the final equilibrium stage of mixing within the mixing volume they do not contain
information about changes in microphysical quantities in the course of mixing.

Pinsky et al. (2016a, hereafter Pt2) analyzed the time evolution of initially monodisperse

and polydisperse DSD during homogeneous mixing. It was shown that result of mixing
strongly depends on the shape of the initial DSD. At a wide DSD, evaporation of droplets
(first of all, of the smallest ones) is not accompanied by a decrease in the mean volume or
effective radius. Moreover, the values of the radii may even increase over time. This result
indicates that the widely used criterion of separation of mixing types based on the behavior of
the mean volume radius during mixing is not generally relevant and may be wrong in
application to real clouds.
Pinsky et.al. (2016b, hereafter Pt3) introduced a diffusion-evaporation model which
describes evolution DSDs and all the microphysical variables due to two simultaneously
occurring processes: turbulent diffusion and droplet evaporation. Mixing between two equal
volumes of subsaturated and cloudy air was analyzed, i.e. it was assumed that the cloud
volume fraction $\mu = 1/2$. The initial DSD in the cloudy volume was assumed monodisperse.
These simplified assumptions allowed to reduce the turbulent mixing equations to two-
parametric ones. The first parameter is the Damkölher number, $Da$, which is the ratio of the
characteristic mixing time to the characteristic  phase relaxation time. The second parameter is
the potential evaporation parameter $R$ characterizing the ratio between the amount of water
vapor needed to saturate the initially dry volume and the amount of available liquid water in
the cloudy volume.
Within the $Da - R$ space, in addition to the two extreme mixing types defined in the
classical concept, two  more  mixing  regimes were distinguished, namely, intermediate and
inhomogeneous mixing. It was shown that any type of mixing leads to formation of a tail of
small droplets, i.e. to DSD broadening. It was also shown that the relative humidity in the
initially dry volume rapidly increases due to both water vapor diffusion and evaporation of
penetrating droplets. As a result, the mean volume and effectice radii in the initially dry
volume rapidly approach the values typical of cloudy volume. At the same time, the liquid
water content (LWC) remains significantly lower than that in the cloudy volume during much
longer time than required for the effective droplet radius to grow.
In the present study (Pt4) we continue investigating the turbulent mixing between an
initially droplet free volume (referred to as dry volume) and a cloudy volume. The focus of
the study is investigation of DSD temporal evolution and analysis of the final equilibrium
DSD. In comparison to Pt3, the problem analyzed in this study is more sophisticated in
several aspects:
• The dependences of different mixing characteristics on cloud volume fraction $0 \leq \mu \leq 1$
are analyzed. In this case the equations of turbulent mixing cannot be reduced to the two-
parametric problem as it was done in Pt3.
• The initial DSDs in cloud volume are polydisperse. We use both narrow and wide
initial DSD described by Gamma distributions with different sets of parameters. The DSD are
the same as those used in Pt2. Mechanisms of formation of wide DSDs in clouds including
DSDs in undiluted cloud cores were investigated in several studies [e.g., Khain et al., 2000;
Pinky and Khain, 2002; Segal et al., 2004; Prabha et al., 2011]. These studies show the DSD
broadening is caused by in-cloud nucleation of droplets within clouds as well as by collisions
between cloud droplets. It was shown that DSDs in adiabatic volumes can be wide and first
raindrops or drizzle drop arise namely in non-diluted adiabatic cloud parcels [Khain et al.,
2013; Magaritz-Ronen et al., 2016]. We use both narrow and wide DSDs in the form of
Gamma distribution with typical parameters used in different cloud resolving models. The
DSDs that are used as initial ones in cloudy volumes could be formed also under influence of
mixing during their previous history. The mechanisms of the formation of initial DSD are not
of interest in the study since that do not affect the analysis.
• The equation for supersaturation, used in this study, is valid at low humidity in the
initially dry volume and is more general and compared with that used in Pt3, which makes the
DSD calculations more accurate.
At the same time, some simplifications used in Pt3 are retained in this study. The vertical
movement of the entire mixing volume is neglected; collisions between droplets and droplet
sedimentation are not allowed. Also, we consider a 1D diffusion-evaporation problem. We
neglect the changes of temperature in the course of mixing, which is possibly a less significant
simplification. All these simplifications allow to reveal the effects of turbulent mixing and
evaporation on DSD evolution.

**2. Formulation of the problem and model design**
In this study, the process of mixing is investigated basing on the solution of 1D diffusion-
evaporation equation (see also Pt3). According to this equation, evaporation of droplets due to
negative supersaturation in the mixing volume takes place simultaneously with turbulent
mixing. Since droplets within the volume are under different negative supersaturation values
until the final equilibrium is reached, the modeled mixing is inhomogeneous. The droplets can
evaporate either partially or totally. The evaporation leads to a decrease in droplet sizes and in
droplet concentration.
Like in Pt3, the process of turbulent diffusion is described by a 1D equation of turbulent
diffusion. The equation does not describe formation of separate turbulent filaments. Instead, it
describes averaged effects of turbulent vortices of different scales  by modeling of turbulent
diffusion, characterized by a typical value of turbulent diffusion coefficient $K$. The mixing is
assumed to be driven by isotropic turbulence at scales within the inertial sub-range where
Richardson's law is valid. In this case, turbulent coefficient is evaluated as in Monin and
Yaglom (1975):
$K(L) = C\varepsilon^{1/3}L^{4/3}$                                                                 (1)
In Eq. (1) $\varepsilon$ is the turbulent kinetic energy dissipation rate and $C = 0.2$ is a constant (Monin
and Yaglom, 1975), Boffetta and Sokolov (2002). Eq. (1) means that we consider the effects
of turbulent diffusion at scales much larger than the Kolmogorov microscale, i.e. the effects of
molecular diffusion are neglected. In the simulations, we use $L = 40\ m$ and $\varepsilon = 20\,cm^2 s^{-3}$. It
means that in the present study mixing is performed by vortices smaller than several tens of
meters which agrees with measurements in warm Cu (Gerber et al. 2008). The value of
turbulent kinetic energy dissipation rate chosen is also typical for small Cu (e.g. Gerber et al.
2008). These parameters correspond to the values of Da of several hundred. The model allows
utilization of other values of  $L$  and  $\varepsilon$  typical of other cloud type (say, deep convective
clouds) which can change results quantitatively, but not qualitatively.

***Geometry of mixing and the initial conditions***
The conceptual scheme presenting mixing geometry and the initial conditions used in the
following analysis are shown in **Figure 1**.

**Fig 1 here**

At $t = 0$ the mixing volume of length $L$ is divided into two volumes: the cloud volume of
length $\mu L$ (Fig.1, left) and the dry volume of length $(1-\mu)L$ (Fig.1, right), where $0 \leq \mu \leq 1$
is the cloud volume fraction. The entire volume is assumed closed, i.e. adiabatic. At $t = 0$ the
cloud volume is assumed saturated, so the supersaturation $S_1 = 0$. This volume is also
characterized by the initial distribution of the square of the droplet radii $g_1(\sigma)$, where $\sigma = r^2$.
The   initial   liquid   water   mixing   ratio   in   the   cloudy   volume   is   equal   to
$q_{w1} = \dfrac{4\pi\rho_w}{3\rho_a}\displaystyle\int_0^\infty \sigma^{3/2} g_1(\sigma)d\sigma$ . The integral of $g_1(\sigma)$ over $\sigma$ is equal to the initial droplet
concentration in the cloud volume $N_1 = \displaystyle\int_0^\infty g_1(\sigma)d\sigma$ . The initial droplet concentration in the
dry volume is $N_2 = 0$, the initial negative supersaturation in this volume is $S_2 < 0$ and the
initial liquid water mixing ratio $q_{w2} = 0$. Therefore, the initial profiles of these quantities
along the $x$-axis are step functions:

$N(x,0) = \begin{cases} N_1 & \text{if} \quad 0 \leq x < \mu L \\ 0 & \text{if} \quad \mu L \leq x < L \end{cases}$                                          (2a)
$S(x,0) = \begin{cases} 0 & \text{if} & 0 \le x < \mu L \\ S_2 & \text{if} & \mu L \le x < L \end{cases}$          (2b)
$q_w(x,0) = \begin{cases} q_{w1} & \text{if} & 0 \le x < \mu L \\ 0 & \text{if} & \mu L \le x < L \end{cases}$          (2c)

The initial profile of droplet concentration is shown in Fig. 1b. This is the simplest
inhomogeneous mixing scheme, wherein mixing takes place only in the $x$-direction, and the
vertical velocity is neglected.
Since the total volume is adiabatic, the fluxes of different quantities through the left and
right boundaries at any time instance are equal to zero, i.e.

$\dfrac{\partial N(0,t)}{\partial x} = \dfrac{\partial N(L,t)}{\partial x} = 0\,;\;\; \dfrac{\partial q_w(0,t)}{\partial x} = \dfrac{\partial q_w(L,t)}{\partial x} = 0\,;\dfrac{\partial q_v(0,t)}{\partial x} = \dfrac{\partial q_v(L,t)}{\partial x} = 0$    (3)
where $q_v$ is the water vapor mixing ratio.
To investigate of mixing process for different initial DSD, we assume that DSD in the cloud
volume can be represented by a Gamma distribution:
$f(r, t = 0) = \dfrac{N_0}{\Gamma(\alpha)\beta} \left( \dfrac{r}{\beta} \right)^{\alpha-1} \exp\left( -\dfrac{r}{\beta} \right)$          (4)
where $N_0$ is an intercept parameter, $\alpha$ is a shape parameter and $\beta$ is a slope parameter of
distribution. The DSD $f(r)$ relates to distribution $g_1(\sigma)$ as $f(r) = 2r g_1(\sigma)$. We performed
simulations with both initially wide and narrow DSDs. The width of DSD is determined by a
set of parameters. The parameters of the initial Gamma distributions used in this study are
presented in **Table 1**. Parameters of the distributions are chosen in such a way that the modal
radii of DSD and the values of LWC are the same for both distributions. These distributions
were used in Pt2 for analysis of homogeneous mixing.

**Table 1 here**


*Conservative quantity* $\Gamma(x,t)$
The supersaturation equation for an adiabatic immovable volume can be written in the
form $\dfrac{1}{S+1}\dfrac{dS}{dt} = -A_2\dfrac{dq_w}{dt}$, where $S$ is supersaturation over water, and the coefficient
$A_2 = \dfrac{1}{q_v} + \dfrac{L_w^2}{c_p R_v T^2}$ is slightly dependent on temperature (Korolev and Mazin, 2003) (notations
of other variables are presented in **Appendix**). In our analysis we consider $A_2$ to be a
constant. As follows from the supersaturation equation, the quantity

$\Gamma(x,t) = \ln\left[S(x,t)+1\right] + A_2 q_w(x,t)$                       (5)

is a conservative quantity, i.e. it is invariant with respect to phase transitions. In Eq. (5),
$|S(x,t)|$ can be comparable with unity by the order of magnitude. The conservative quantity
$\Gamma(x,t)$ obeys the following equation for turbulent diffusion

$\dfrac{\partial \Gamma(x,t)}{\partial t} = K \dfrac{\partial^2 \Gamma(x,t)}{\partial x^2}$                       (6)

with the adiabatic (no flux) condition at the left and right boundaries $\dfrac{\partial \Gamma(0,t)}{\partial x} = \dfrac{\partial \Gamma(L,t)}{\partial x} = 0$
and the initial profile at $t = 0$

$\Gamma(x,0) = \begin{cases} A_2 q_{w1} & \text{if} & 0 \le x < \mu L \\ \ln\left[S_2 +1\right] & \text{if} & \mu L \le x < L \end{cases}$            (7)

From Eq. (7) it follows that $\Gamma(x,0)$ is positive in the cloud volume and negative in the
initially dry volume. The mean value of function $\Gamma(x,0)$ can be written as follows:

$$\bar{\Gamma} = \frac{1}{L}\int_0^L \Gamma(x,0)dx = \frac{A_2 q_{w1}}{L}\int_0^{\mu L} dx + \frac{\ln[S_2+1]}{L}\int_{\mu L}^L dx = \mu A_2 q_{w1} + (1-\mu)\ln[S_2+1] \quad (8)$$

$\bar{\Gamma}$ can be either positive or negative. In the latter case a complete evaporation of droplets in the
course of mixing takes place.

The solution of Eq. (6) with the initial condition (7) is (Polyanin et al., 2004):

$$\Gamma(x,t) = \sum_{n=0}^{\infty} a_n \exp\left(-\frac{Kn^2\pi^2 t}{L^2}\right)\cos\left(\frac{n\pi x}{L}\right) =$$
$$\mu A_2 q_{w1} + (1-\mu)\ln[S_2+1] - \qquad\qquad (9)$$
$$2\left(\ln[S_2+1] - A_2 q_{w1}\right)\sum_{n=1}^{\infty}\frac{\sin(n\pi\mu)}{n\pi}\exp\left(-\frac{Kn^2\pi^2 t}{L^2}\right)\cos\left(\frac{n\pi x}{L}\right)$$

One can see that function $\Gamma(x,t)$ depends on three independent parameters $A_2 q_{w1}$, $S_2$ and $\mu$.
This function does not depend on the shape of the initial DSD in the cloud volume. In the final
state when $t \to \infty$, $\Gamma(x,t)$ is :
$$\Gamma(t=\infty) = \mu A_2 q_{w1} + (1-\mu)\ln[S_2+1] \qquad\qquad (10)$$
Therefore, $\Gamma(t=\infty)$ depends on the cloud fraction and the initial values of liquid water
mixing ratio in the cloud volume and the relative humidity in initially dry volume.

The final equilibrium values of supersaturation $S(x,\infty)$ and liquid water mixing ratio

$q_w(x,\infty)$ can be calculated using Eq. (5). The case $\Gamma(t=\infty) > 0$ corresponds to the
equilibrium state with $S(x,\infty) = 0$ and $q_w(x,\infty) = \mu q_{w1} + (1-\mu)\frac{\ln[S_2+1]}{A_2}$, when droplets
remain, but do not evaporate any longer.
The case $\Gamma(t=\infty)<0$ corresponds to the equilibrium state with $q_w(x,\infty)=0$ and
$S(x,\infty)=(1+S_2)^{1-\mu}\exp(\mu A_2 q_{w1})-1$. In this equilibrium state droplets are totally evaporated,
and volume remains subsaturated $S(x,\infty)<0$. At given $q_{w1}$ and $S_2$, there is a critical value of
the cloud fraction $\mu_{cr}$ which separates these two possible final equilibrium states. This critical
value corresponds to $\Gamma(t=\infty)=0$ and can be calculated from Eq. (10) as:

$$\mu_{cr}=\frac{\ln[S_2+1]}{\ln[S_2+1]-A_2 q_{w1}} \qquad (11)$$

Another expression for $\mu_{cr}$ was formulated in Pt1.
The examples of spatial-temporal variations of function $\Gamma(x,t)$ for different cloud
fractions and initial RH=80% are shown in **Figure 2.**

**Fig 2 here**

Upper panels $\mu=0.1$ correspond to the case of final total droplet evaporation and negative
final function $\Gamma$, whereas the middle and bottom rows $\mu=0.5$ and $\mu=0.9$ illustrate partial
evaporation cases when the total mixing volume reaches saturation. It is interesting that the
time required for the final equilibrium state to be reached practically does not depend on the
cloud fraction, being ~180 seconds for the illustrated cases. The cases $\mu=0.1$ and $\mu=0.9$
demonstrate a strong non-symmetric spatial variability of $\Gamma(x)$ function during the first 50
seconds. At $\mu=0.5$, a nearly full compensation between saturation deficit in the dry volume
and available liquid water in the cloud volume takes place  if at the equilibrium state
$S(x,\infty)=q_w(x,\infty)=\Gamma(x,\infty)=0$. However, the compensation at $\mu=0.5$ is not full because of
the nonlinearity of $\Gamma$ in Eq. (5).

*Diffusion-evaporation equation for DSD*
To formulate the diffusion-evaporation equation we use a simplified equation for droplet
evaporation (Pruppacher and Klett, 1997), in which the curvature term and the chemical
composition term are omitted

$$\frac{d\sigma}{dt} = \frac{2S}{F} \tag{12}$$

where $F = \frac{\rho_w L_w^2}{k_a R_v T^2} + \frac{\rho_w R_v T}{e_w(T)D} = const$ (Notations of other variables are presented in Appendix.)
The solution of Eq. (12) is

$$\sigma(t) = \frac{2}{F} \int_0^t S(t')dt' + \sigma_0 \tag{13}$$

Eq. (13) means that in the course of evaporation, distribution $g(\sigma)$ shifts to the left without
changing its shape. The diffusion-evaporation equation for function $g(x,t,\sigma)$ can be written
in the form

$$\frac{\partial g}{\partial t} = K \frac{\partial^2 g}{\partial x^2} + \frac{\partial}{\partial \sigma}\left(\frac{d\sigma}{dt}g\right) \tag{14}$$

Combining Eqs. (12) and (14) yields

$$\frac{\partial g(x,t,\sigma)}{\partial t} = K \frac{\partial^2 g(x,t,\sigma)}{\partial x^2} + \frac{2S}{F}\frac{\partial g(x,t,\sigma)}{\partial \sigma} \tag{15}$$


Eq. (15) is similar to the diffusion-evaporation equation for size distribution function used in
Pt 3. The first term on the right hand side of Eq. (15) describes the effect of turbulent
diffusion, while the second term describes the changes of size distribution due to droplet
evaporation. To close this equation, one can use Eq. (5) written as

$$S(x,t) = \exp\left[\Gamma(x,t) - A_2 q_w(x,t)\right] - 1,$$ (16)

and the equation for liquid water mixing ratio

$$q_w(x,t) = \frac{4\pi\rho_w}{3\rho_a} \int_0^\infty \sigma^{3/2} g(x,t,\sigma) d\sigma$$ (17)
The equation system (15-17) for distribution $g(x,t,\sigma)$ should be solved under the following
initial condition
$$g(x,0,\sigma) = \begin{cases} g_1(\sigma) & \text{if} & 0 \le x < \mu L \\ 0 & \text{if} & \mu L \le x < L \end{cases}$$ (18)
and using the Neumann boundary conditions

$$\frac{\partial g(0,t,\sigma)}{\partial x} = \frac{\partial g(L,t,\sigma)}{\partial x} = 0$$ (19)

These equations were solved numerically on a linear grid of droplet radii $r_j$ being within
the range 0-50 μm, where $j = 1...50$ are the bin numbers. The number of grid points along the
$x$-axis was set equal to 81. In numerical calculations, the "evaporation term" in Eq. (15) was
approximated as
$$\frac{2S}{F} \frac{\partial g(x,t,\sigma)}{\partial \sigma} \approx \frac{g\left(x,t,\sigma + \frac{2S}{F}\Delta t\right) - g(x,t,\sigma)}{\Delta t}.$$ (20)

A shift and subsequent remapping of DSD using the method proposed by Kovetz and Olund's
(1969) were implemented to solve Eq. (20) with the help of MATLAB solver PDEPE. After
calculation of $g(x,t,\sigma_j)$ function, DSD $f(x,t,r_j)$ was calculated using the relationship
$$f(x,t,r_j) = 2r_j g(x,t,\sigma_j).$$

## 3. Spatial-temporal variations of DSD and of DSD parameters


Mixing may take a significant time. Cloud microphysical parameters measured in *in-situ*
observations correspond to different stages of this transient mixing process. During mixing,
DSDs and its parameters change substantially, which makes it reasonable to analyze these
time changes.
**Figure 3** shows time evolution of initially narrow DSD in the centers of the cloudy volume
and of the initially dry volume. The values of DSD in the initially cloudy volume decrease
while there are no significant changes in the DSD shape. At $\mu = 0.7$, the droplet radius
corresponding to the DSD maximum remains unchanged during mixing staying equal to 10
$\mu m$. At $\mu = 0.3$ the effect of droplet diffusion on DSD is stronger, and mixing leads not
only to a  decrease in the DSD values, but also to a decrease in the peak droplet radius in the
cloudy volume. Both at $\mu = 0.3$ and $\mu = 0.7$, mixing leads to broadening of the initial DSD
due to the appearance of the tail of small droplets. The tail of small droplets is especially
pronounced in the initially dry volume since maximum evaporation of penetrated droplets.
The rate of the DSD growth in the initially dry volume, depends on the value of the cloud
fraction. At a low cloud fraction, DSD maximum (i.e. drop concentration and drop mass)
remains substantially lower for the most period of mixing process than that in the cloudy
volume. At the same time, the radius corresponding to the DSD maximum increases reaching
80% of its maximum value already within the first 5 s. This is due to the fast increase in the
relative humidity during mixing, so large droplets penetrating the initially dry volume do not
decrease in size anyhow significantly determining the values of modal, mean volume and
effective radii. Thus, we see two stages of DSD evolution within in the initially dry volumes:
at the first stage penetrated droplets evaporate totally or partially forming the tail of small
droplets. The formation of the tail of smallest droplets does not lead to a significant changes
of the size of the largest droplets.   Note that according to equation of diffusion
growth/evaporation in of sub-saturation conditions, the rate of droplet radii decreases inverse
proportionally to the droplet radius. It means that if, say, radius of a 2 $\mu m$ droplet decreases
twice during a certain time instance, the radius of 20 $\mu m$ droplet will decrease by less than
0.1 $\mu m$, i.e. remains approximately unchanged. At this stage diffusion of water vapor from
cloudy volume and evaporation of penetrating droplets lead to a rapid growth of relative
humidity RH. This growth of RH decreases evaporation rate of droplets penetrating initially
dry volume later. At the second stage mixing leads to the increase in the droplet number due
to droplet diffusion from cloudy volume. Since, RH is high, this diffusion is not accompanied
by significant change droplet sizes, so DSD grows similarly at all radii.

**Figure 3 here**

At the initially wide DSD (**Figure 4**), the radii of the DSD maximum do not change. It
means that at the initial RH= 80%, mixing and evaporation lead to a fast saturation of the
initially dry volume, after which the peak radius remains unchanged in this volume. In the
initially cloud volume RH remains close to 100% so the DSD decrease is related to dilution by
the air from initially dry volume.

**Figure 4 here**

It is interesting that at $\mu = 0.3$ in the initially dry volume, DSD reaches its maximum during
the transition period (Fig.4, at t=80s), and then decreases toward the equilibrium state. This
behavior is caused by the competition between the diffusion and droplet evaporation.
**Figure 5** shows spatial dependences of droplet concentration, LWC and the mean volume
radius within the mixing volume at different time instances at narrow initial DSD.  At small
values of the cloud fraction, diffusion of water vapor and droplets, as well as droplet
evaporation lead to a fast decrease in droplet concentration and in LWC in the initially cloud
volume. The mean volume radius in this volume decreases by about 15% in the course of
mixing. It is natural that at large cloud fraction, droplet concentration and LWC in the initially
cloudy volume decrease slowly, while these quantities in the initially dry volume increase
rapidly. At both small and large cloud fractions, the mean volume radius in the initially dry
volume grows rapidly during the mixing toward its values in the initially cloudy volumes,
even if droplet concentration and LWC remain much lower than in the adjacent cloud volume.

**Figure 5 here**

**Figure 6** shows the spatial dependences of droplet concentration, LWC and the mean
volume radius within the mixing volume at different time instances at wide initial DSD.


**Figure 6 here**


A specific feature of mixing at  a wide DSD is the increase in the mean volume radius, so the
ratio $\frac{r_v}{r_{v0}} > 1$. In the course of mixing, the mean volume radius maximum is reached in the
initially dry volumes. This result can be attributed to the fact that in this volume smaller
droplets fully evaporate, so the concentration of large droplets increases with respect to
concentration of smaller droplets (Fig. 4, right column). Scattering diagrams plotted using *in-*
*situ* observations often contain points or groups of points with $\frac{r_v}{r_{v0}} > 1$ (or $\frac{r_e}{r_{e0}} > 1$ , where $r_e$ is
effective radius) within wide range of normalized droplet concentration (e.g., Burnet and
Brenguier, 2007; Krueger et al., 2006, Gerber et al., 2008). The result obtained in the present
study shows that the behavior of $\dfrac{r_v}{r_{v0}}$ with time in the course of mixing may depend of the
DSD shape in the initially cloud volume that determines relationship between concentrations
of small and large droplets in course of mixing. Of course, the DSD shape is only one possible
reason of appearance of points with $\dfrac{r_v}{r_{v0}} > 1$ on the scattering diagram.
We see that the transition to the final equilibrium state within the volume with the spatial
scale of 40 m is about 5 min (Fig. 8), which is a comparatively long period of time compared
to the characteristic times of other microphysical processes, including droplet evaporation.
During this time the DSD changes substantially, especially at small cloud fraction. The mean
volume radius in the initially dry volume increases much faster than LWC. As a result, mean
volume radius in such volume rapidly reaches the values typical of cloudy air, while LWC
still remains substantially lower than in the cloudy volume. Despite some DSD broadening,
the final DSDs in the mixing volume resemble those in the initially cloud volumes. The main
effect of mixing is lowering the DSD values as the cloud fraction decreases.

**4. Equilibrium state and mixing diagram**
This study reconsiders the classical theory of mixing diagrams. In the classical theory
two volumes (cloudy and droplet free) mix with each other within a given unmovable mixing
volume (see review by Korolev et al., 2016). Mixing diagrams are typically plotted for times
when all variables become uniform within the mixing volume, i.e when the equilibrium state
is reached. We plot the mixing diagram using the same simplifications used in the plotting
classical mixing diagrams, namely: no vertical motions and no collisions are assumed. These
assumptions allow to reveal better the microphysical effects of turbulent mixing. It is widely
assumed that the mixing type is determined by the Damkohler number that depends only on
drop relaxation time and mixing time. No averaged vertical velocity and no collision rate are
included into this criterion.
We extend the theory, however, in several important aspects concerning microphysical
effects: a) we consider time dependent process of mixing and b) initial droplet size
distributions are assumed polydisperse.
Mixing considered in the present study always leads to the equilibrium state. As was
explained above, two equilibrium states are possible. The first one is characterized by the total
evaporation of cloud droplets $q_w(x,\infty)=0$, whereas the second one occurs if the air in the
mixing volume becomes saturated, i.e. when $S(x,\infty)=0$. At the given initial values of $q_{w1}$ in
the cloud volume and of $S_2$ in the initially dry volume, there always exists the cloud fraction
$\mu_{cr}$ (Eq. 11) separating these two states.

**4.1. The process of achieving the equilibrium state**
**Figure 7** shows the dependences of the time required to reach the equilibrium on the cloud
fraction, at different initial relative humidity values in the dry volume and two initial DSDs
(the parameters are presented in Tab.1). The characteristic time is defined here as the time
from the beginning of mixing to the time instance when inequality $\delta = \dfrac{\bar{N}(t)-\bar{N}(\infty)}{\bar{N}(0)-\bar{N}(\infty)} < 0.01$
becomes valid. The mean droplet concentration is calculated by averaging along $x$-axes
($\bar{N}(t)=\dfrac{1}{L}\int\limits_0^L N(x,t)dx$). In case of a total evaporation, $\bar{N}(\infty)=0$.

**Figure 7 here**

Each curve in Fig. 7 consists of two branches. The left branches correspond to the total
evaporation regime, while the right branches correspond to the partial evaporation at
equilibrium. The maximum time corresponds to the situation when the available amount of
liquid water is approximately equal to the saturation deficit. A similar result was obtained in
Pt1 and Pt2 for homogeneous mixing. The maximum values of the characteristic time are
about 4 min for a mixing volume of 40 m in length. The right branches show that the
characteristic time decreases with increasing cloud fraction. Despite some differences in the
curve slopes, the characteristic times for wide and narrow DSD are quite similar.

**Figure 8** shows dependences of normalized cube of the mean volume radius on the cloud

fraction at different time instances for two values of $x$: $x = 0$ (solid lines) corresponds to the
initially cloudy volume, and $x = L$ (dashed line) corresponds to the initially dry volume. The
figure is plotted for the narrow DSD for two values of $RH_2$: 60% and 95%. Despite the fact
that the diffusion-evaporation equation allows simulating using any initial RH, we do not
consider in our examples the cases of very low RH of dry volume. It is because at very low
RH, say, RH=20%, the cloud fraction should exceed 0.8 to prevent total droplet evaporation
in the equilibrium state (at LWC=1 g/kg). At the same time, we are interested in the
equilibrium state at which droplets exist. Note that at the lateral edges of warm Cu a shell of
humid air arises around cloud, so RH of the entrained air should be high enough (e.g. Gerber
et al., 2008).

**Figure 8 here**


The curve plotted for the time instance of 300 s corresponds to the equilibrium state (hereafter
the equilibrium curve). The curves above the equilibrium curve correspond to the initially
cloudy volume, and the curves below the equilibrium curve correspond to the initially dry
volume. One can see how curves of both types approach the same final state. During the
mixing the curves move over the $\left(\dfrac{r_v}{r_{v0}}\right)^3 - \mu$ plane toward the equilibrium curve. As a result,
the curves plotted in Fig.8, corresponding to different time instances of the mixing, together
cover the entire area of the panels.
During this movement the distance from the curves to the horizontal line $\left(\dfrac{r_v}{r_{v0}}\right)^3 = 1$ changes,
and the curves slopes increase. In our case of $L = 40\,\text{m}$, the mixing remains inhomogeneous
the during entire mixing process, so the change in the distance from the curves to the
horizontal line $\left(\dfrac{r_v}{r_{v0}}\right)^3 = 1$ characterizes the temporal changes over the mixing process, but not
a change in mixing type.
It is noteworthy in this relation that scattering diagrams plotted using *in-situ* observations
reflect mixing between different multiple volumes at different stages of the mixing process.
Accordingly, points in the scattering diagrams can be far from the equilibrium location. Fig. 8
indicates, therefore, that scattering diagrams show snapshots of transient mixing process when
the distance from points in the diagrams to line $\left(\dfrac{r_v}{r_{v0}}\right)^3 = 1$ characterize the stage of the
mixing process, but not the mixing type.
The dependences of normalized cube of the mean volume radius on the cloud fraction at
different time instances at wide DSD also indicate approaching to the equilibrium curve,
while all the curves correspond to $\left(\dfrac{r_v}{r_{v0}}\right)^3 > 1$ (not shown).
Note that in several studies normalized effective radius is used for plotting scattering and
mixing diagrams, but not mean volume radius (Gerber et al. 2008; Freud et al., 2011).
Comparison of scattering and mixing diagrams in the study plotted using mean volume and
effective radii did not reveal any significant differences (not shown).

**4.2. Mixing diagrams**
Using the diffusion-evaporation equations (15-17) we calculated the equilibrium DSD for
different initial relative humidity values and different cloud fractions. Each calculation was
performed for both narrow and wide initial DSD (parameters shown in Tab.1). These
equilibrium DSD were used to calculate mixing diagrams showing dependences of normalized
cube of the effective radius on the cloud fraction.
The corresponding mixing diagrams for homogeneous mixing case were also calculated
for comparison. To this effect, the supersaturation and DSD in both the cloud and the dry
volumes were aligned, taking into account the cloud fraction value $\mu$. The alignment led to
the following initial values of supersaturation and DSD within the mixing volume:

$\quad S_0 = (1-\mu)S_2; \quad g_0(\sigma) = \mu g_1(\sigma)$ (21)

Upon the alignment, time evolution values of DSD under homogeneous evaporation in an
adiabatic immovable parcel were calculated until the equilibrium state was reached. These
equilibrium DSD were used to calculate mixing diagrams for homogeneous mixing. To do
this, we used the parcel model proposed by Korolev (1995) that describes evaporation by
means of equations with temperature-dependent parameters. **Figure 9** shows the mixing
diagrams plotted for initial narrow and wide DSD cases.

**Figure 9 here**


While all the curves in the mixing diagram for narrow DSD are below the straight line
$\left(\dfrac{r_v}{r_{v0}}\right)^3 = 1$, the curves for wide DSD are above this line. The explanation of this effect is given
in Section 3 (Fig. 6). The curves plotted for homogeneous and inhomogeneous mixing
demonstrate an important feature. Namely, at given values of RH and $q_{w1}$ in the initially dry
volume, the values $\mu_{cr}$ of the cloud fraction at which all the droplets evaporate are
approximately the same for any type of mixing. This condition is the consequence of the mass
conservation law determined by Eq. (11) and does not depend of the initial DSD shape. In
standard mixing diagrams (e.g. Lehmann et al., 2009; Gerber et al., 2008; Freud et al., 2011),
the horizontal straight line $\left(\dfrac{r_v}{r_{v0}}\right)^3 = 1$ (or $\left(\dfrac{r_e}{r_{e0}}\right)^3 = 1$) is typically plotted for the entire range of
the cloud fraction [0...1], while the curves corresponding to homogeneous mixing are plotted
for different RH within the range $[\mu_{cr}(RH_2)...1]$. As a result, the high difference between
extremely inhomogeneous and homogeneous mixing types is clearly seen at low RH and at
small cloud fractions. The condition that $\mu_{cr}$ is the same for different mixing types indicates
that the mixing diagrams may look nearly similar for $\mu > \mu_{cr}$. It means that the range of the
cloud fractions required for comparison of diagrams aimed at determination of a mixing type
shortens as $RH_2$ values in the surrounding air decrease.
The comparison of the left and the right panels in Fig. 9 shows that the differences
between the diagrams for homogeneous and inhomogeneous mixing types are more
pronounced for initially narrow DSD. The maximum difference should take place for
monodisperse DSD considered in in Pt1, Pt2 and Pt3. Within the range of $\mu > \mu_{cr}$, the
distance between the curves corresponding to different mixing regimes is small even for
narrow DSD and low $RH_2$. The lower difference is related to the fact that at high $RH_2$ the
curves in the mixing diagrams are close to the horizontal straight line in both regimes, while at
low $RH_2$, $\mu_{cr}$ is small and both curves should drop to zero in the vicinity of $\mu = \mu_{cr}$.
As regards the wide DSD case, the difference between the curves corresponding to
different mixing type is negligible (Fig. 9, right)

**4.3. Effect of the relative humidity**
In measurements carried out at cloud boundaries and in cloud simulations, the cloud
fraction is not known, therefore it is widely accepted to use normalized droplet concentration
instead of the cloud fraction (Burnet and Brenguier, 2007; Gerber et al., 2008: Lehmann et al.,
2009). Droplet concentration is normalized by the maximum value along the airplane traverse.
The difference between the cloud fraction and normalized droplet concentration is obvious:
the cloud fraction is a parameter given as the initial condition. At the same time, normalized
droplet concentration changes with time and space due to complete evaporation of some
droplet fraction. **Figure 10** shows dependencies of normalized droplet concentration on the
cloud fraction at the equilibrium final state of mixing. One can see a substantial deviation
from 1:1 linear dependence, especially at low RH. As we know, droplet concentration
decreases in the course of both homogeneous and inhomogeneous mixing if the initial DSD
are polydisperse. The fraction of totally evaporating droplets increases with decreasing $RH_2$.
As expected, droplet concentration in homogeneous mixing is higher than that in
inhomogeneous mixing. The difference between droplet concentrations at wide DSD is lower
than at narrow DSD.

**Fig. 10 here**

**Figure 11** shows the dependencies $\left( \dfrac{r_v}{r_{v0}} \right)^3$ on normalized droplet concentration for narrow
and wide DSD in inhomogeneous mixing. The normalization by droplet concentration in the
initially cloud volume at $t = 0$ was used. Taking into account the dependences of normalized
droplet concentration on the cloud fraction $\mu$ (Fig. 10), one can get the curves shown in Fig.
11 which actually coincide at different $RH_2$. The lack of the sensitivity to $RH_2$ can be
attributed to the fact that a decrease in RH leads to a decrease in normalized droplet
concentration, so the curves corresponding to low RH in Fig. 9 shift to the left when the
normalized droplet concentration is used instead of $\mu$. The shape of the dependences in Fig
11 (right) is explained by an increase in the mean volume radius with decreasing droplet
concentration.

**Fig 11 here**

Thus, the mixing diagrams plotted in the plane $\left(\dfrac{r_v}{r_{v0}}\right)^3$ vs normalized droplet
concentration do not depend on the relative humidity of the surrounding dry air. This result
indicates an additional difficulty in distinguishing between mixing types based on scattering
diagrams plotted using *in-situ* data in these axes. The concentration of observed points in
these scattering diagrams close to the line $\left(\dfrac{r_v}{r_{v0}}\right)^3 = 1$ is often interpreted as an indication of
homogeneous mixing, but at high RH in the surrounding air (Gerber et al., 2008; Lehmann et
al., 2009). High values of RH in the penetrating air volumes are usually explained by
formation of a layer of most air around the cloud boundary (Gerber et al., 2008, Knight and
Miller, 1998).
The reference values of droplet concentration and the effective radius used for
normalization in the present study are taken as the initial values in the cloud volume before it
mixes with the neighbouring dry volume. In real *in-situ* measurements the reference values of
these quantities are typically chosen in a less diluted cloud volume along the airplane traverse.
This reference volume may be quite remote from the particular mixing volume. It can lead to
a shift of the mixing diagram with respect to the $\left(\dfrac{r_v}{r_{v0}}\right)^3 = 1$ line, as well as to a large variation
in mixing diagram shapes, unrelated, however, to the mixing type (e.g., Lehmann et al.,

2009).


## 5.     Discussion and conclusion


This study extends the analysis of mixing performed in Pt3 where the diffusion-
evaporation equation served as the basis, the initial DSD were assumed monodisperse and
the cloud fraction was chosen as $\mu = 1/2$. In the present study, the analysis focuses on the
temporal and spatial evolution of initially polidisperse DSD and investigates mixing diagrams
obtained for narrow and wide initial DSD within a wide range of the cloud fraction values (0.1
- 0.95). It is shown that results of mixing and the structure of mixing diagrams depend on the
initial DSD shape. This finding indicates that mixing is a multi-parametrical problem that
cannot be determined by a single parameter (e.g. the Damkölher number as often assumed) or
even by two parameters (the Damkölher number and the potential evaporation parameters as
assumed in Pt3). The temporal changes of DSD and their moments during mixing are
calculated. Although DSD broaden, they tend to remain similar to the original DSD. The main
changes come from the cloud air dilution by the dry air, which leads to a decrease in droplet
concentration for all droplet sizes. The changes of DSD and its shape are minimum in the
initially cloud volumes, especially at significant cloud fractions. The droplet radii
corresponding to the DSD peak do not change anyhow significantly. In the initially dry
volumes, mixing and evaporation of penetrated droplets leads to a rapid increase in RH.
Consequently, large droplets penetrating these volumes do not change their sizes significantly.
As a result, the mean volume radius in these volumes rapidly increases and reaches the values
typical of cloud volumes, while LWC remains lower than in the cloud volume for most of the
mixing time. At narrow DSD, the mean volume (and effective) radius remains smaller than
that in the initially cloud volume. At wide DSD, the mean volume (and effective) radius may
become larger than that in the initial DSD. This increase in the effective radius is attributed to
the fact that evaporation of smaller droplets leads to the increase in the fraction of larger
droplets in the DSD. In this study, and in Pt3 it is shown that mixing leads to DSD
broadening. This contrasts with the classical theory, when initially monodisperse DSDs
remain monodisperse in course of mixing. This problem is analyzed in detail in Pt 3. Note
that in real clouds DSD there are many mechanism leading to DSDs broadening (e.g. Pinsky
and Khain 2002).

Dependences of normalized cube of the mean volume radius on the cloud fraction

$\left( r_v / r_{v0} \right)^3$ as a function of $\mu$ at different time instances form the set of curves filling the entire
$\left( r_v / r_{v0} \right)^3 - \mu$ plane. Therefore, both the slope and the distance of these curves in respect to
the horizontal line $\left( r_v / r_{v0} \right)^3 = 1$ change with time. It means that this distance characterizes
the temporal changes in the course of mixing, but not the mixing type (which remains
inhomogeneous during the entire mixing time). The mixing process is comparatively long
(several minutes), so the final equilibrium stage is hardly achievable in real clouds.

It is highly significant that the critical values of the cloud fraction $\mu_{cr}$ corresponding to

total droplet evaporation are the same for any mixing type. It means that the curves in a
mixing diagram corresponding to homogeneous and inhomogeneous mixing types should be
compared only within the range of $\mu > \mu_{cr}$. The range width of $\mu > \mu_{cr}$ decreases with
decreasing relative humidity in the initially dry volume. Taking into account significant
scattering of observed points, this condition greatly hampers the problem of how to
distinguish between mixing types,

Another important result of the study is that mixing diagrams for homogeneous and

inhomogeneous mixing plotted for polydisperse DSD do not differ much. The largest
difference takes place for initially narrow DSD (the maximum difference takes place for
initially monodisperse DSD), but even in this case the difference is not large enough to
reliably distinguish mixing type, owing the significant scatter of observed data. At wide DSD,
this difference between mixing diagrams for homogeneous and inhomogeneous becomes
negligibly small.
The cloud fraction $\mu$ is a predefined parameter and is not determined from observations.
Consequently, in the analysis of *in-situ* measurements the normalized droplet concentration is
typically used instead of the cloud fraction. However, there is a significant difference
between the cloud fraction prescribed a priori and the normalized droplet concentration that
changes due to total evaporation of some fraction of droplets. We have shown that the
utilization of normalized droplet concentration in mixing diagrams is not equivalent to the
utilization of the cloud fraction. The important conclusion is that when mixing diagrams are
plotted using the normalized concentration, the sensitivity to RH disappears. This conclusion
is valid even when RH in the initially dry volume is as low as 60%. This conclusion clearly
contradicts the wide-spread assumption that mixing types can be easily distinguished in
mixing diagrams in case of low relative humidity of the surrounding air.
In the present study as well as in Pt3 and LES performed by Andrejczuk et al. (2006, 2009),
Khain et al. (2017) it is shown that time needed to establishing of equilibrium either quite long or
even never reached. It means that the scattering diagrams observed in situ are just snapshots of
the transient mixing process. However, since the classic mixing diagrams are plotted namely for
equilibrium states, we investigate the transition to such equilibrium assuming that the mixing
volume remains adiabatic (i.e. isolated) during the entire period of mixing. This is, of course, a
serious simplification made to compare the results with those predicted by classical concept.
To sum up, our general conclusion is that the simplifications underlying the classical
concept of mixing are too crude, making it impossible to use mixing diagrams for
comprehensive analysis of mixing and especially for determination of mixing types. At the
same time, mixing diagrams may contain useful information concerning the DSD width.

*Acknowledgements*
This research was supported by the Israel Science Foundation (grants 1393/14, 2027/17)
and the Office of Science (BER) of the US Department of Energy (Award DE-SC0006788,
DE-FOA-0001638). Codes of the diffusional-evaporation model are available upon request.

.




**Appendix. List of symbols**

| Symbol | Description | Units |
|--------|-------------|-------|
| $A_2$ | $\dfrac{1}{q_v} + \dfrac{L_w^2}{c_p R_v T^2}$ , coefficient | - |
| $a_n$ | Fourier series coefficients | - |
| $C$ | Richardson's law constant | - |
| $c_p$ | specific heat capacity of moist air at constant pressure | $\text{J kg}^{-1}\text{K}^{-1}$ |
| $D$ | coefficient of water vapor diffusion in air | $\text{m}^2\text{ s}^{-1}$ |
| $Da$ | *Damkölher* number | - |
| $e$ | water vapor pressure | $\text{N m}^{-2}$ |
| $e_w$ | saturation vapor pressure above flat surface of water | $\text{N m}^{-2}$ |
| $F$ | $\left( \dfrac{\rho_w L_w^2}{k_a R_v T^2} + \dfrac{\rho_w R_v T}{e_w(T) D} \right)$ , coefficient | $\text{m}^{-2}\text{ s}$ |
| $f(r)$ | droplet size distribution | $\text{m}^{-4}$ |
| $g(r)$ | droplet size distribution | $\text{m}^{-5}$ |
| $g_0(\sigma)$ | initial distribution of square radius in homogeneous mixing | $\text{m}^{-5}$ |
| $g_1(\sigma)$ | initial distribution of square radius | $\text{m}^{-5}$ |
| $k_a$ | coefficient of air heat conductivity | $\text{J m}^{-1}\text{s}^{-1}\text{K}^{-1}$ |

| $K$ | turbulent diffusion coefficient | $m^2 s^{-1}$ |
|---|---|---|
| $L$ | characteristic spatial scale of mixing | m |
| $L_w$ | latent heat for liquid water | $J\,kg^{-1}$ |
| $N$ | droplet concentration | $m^{-3}$ |
| $N_0$ | Parameter of Gamma distribution | $m^{-3}$ |
| $\bar{N}$ | mean droplet concentration | $m^{-3}$ |
| $N_1$ | initial droplet concentration in cloud volume | $m^{-3}$ |
| $p$ | pressure of moist air | $N\,m^{-2}$ |
| $q_v$ | water vapor mixing ratio (mass of water vapor per 1 kg of dry air) | - |
| $q_w$ | liquid water mixing ratio (mass of liquid water per 1 kg of dry air) | - |
| $q_{w1}$ | liquid water mixing ratio in cloud volume | - |
| $R$ | $\dfrac{S_2}{A_2 q_{w1}}$ , non-dimensional parameter | - |
| $R_a$ | specific gas constant of moist air | $J\,kg^{-1}K^{-1}$ |
| $R_v$ | specific gas constant of water vapor | $J\,kg^{-1}K^{-1}$ |
| $r$ | droplet radius | m |
| $r_1$ | initial droplet radius | m |
| $r_e$ | effective radius | m |
| $r_{e0}$ | initial effective radius | m |
| $S$ | $e/e_w - 1$, supersaturation over water | - |
| $S_2$ | initial supersaturation in the dry volume | - |
| $S_0$ | initial supersaturation in homogeneous mixing | - |

| | | |
|---|---|---|
| $T$ | temperature | K |
| $t$ | time | s |
| $x$ | distance | m |
| $\alpha$ | parameter of Gamma distribution | - |
| $\beta$ | parameter of Gamma distribution | $m^{-1}$ |
| $\Delta t$ | time step | s |
| $\mu$ | cloud fraction | - |
| $\mu_{cr}$ | critical cloud fraction | - |
| $\varepsilon$ | turbulent dissipation rate | $m^2s^{-3}$ |
| $\Gamma(x,t)$ | conservative function | - |
| $\rho_a$ | air density | kg m$^{-3}$ |
| $\rho_w$ | liquid water density | kg m$^{-3}$ |
| $\sigma$ | square of droplet radius | $m^2$ |

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

**Tab.1** Parameters of the initial Gamma distributions

| DSD | $N_0$, cm$^3$ | $\alpha$ | $\beta$, μm | Modal radius, μm | LWC, g/m$^3$ |
|---|---|---|---|---|---|
| Narrow | 264.2 | 101.0 | 0.1 | 10.0 | 1.178 |
| Wide | 71.0 | 4.3 | 3.1 | 10.0 | 1.178 |
































**Fig.1.** The initial state at $t = 0$. The left volume is a saturated cloudy volume; the right
volume is an under-saturated dry air volume.






























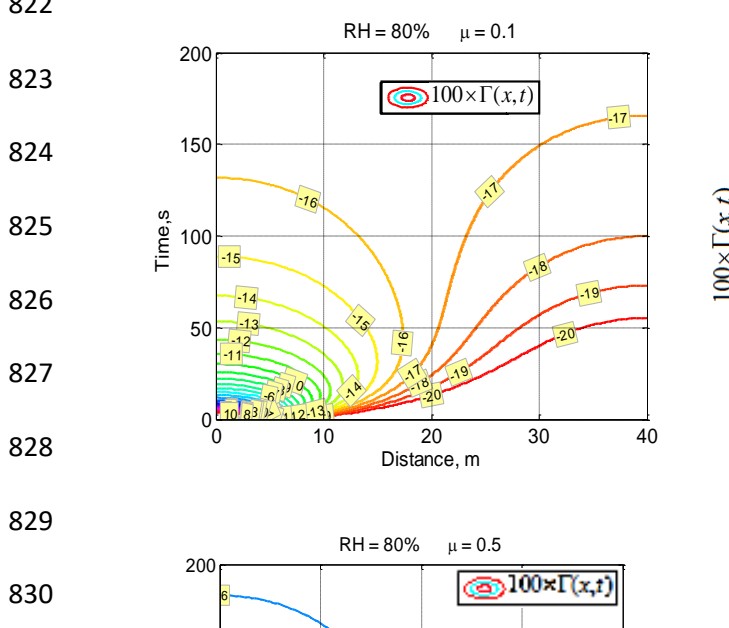
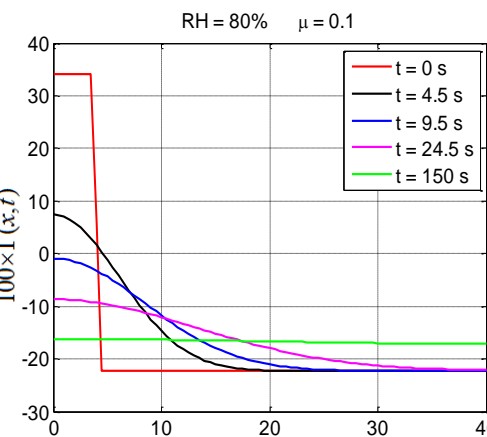
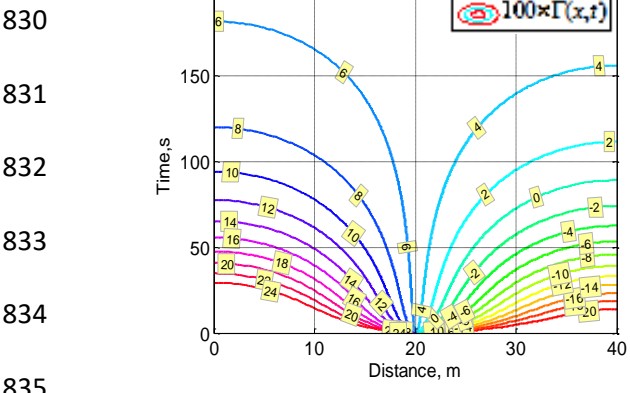
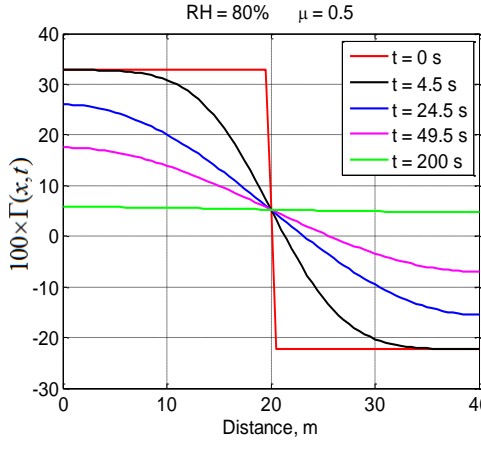
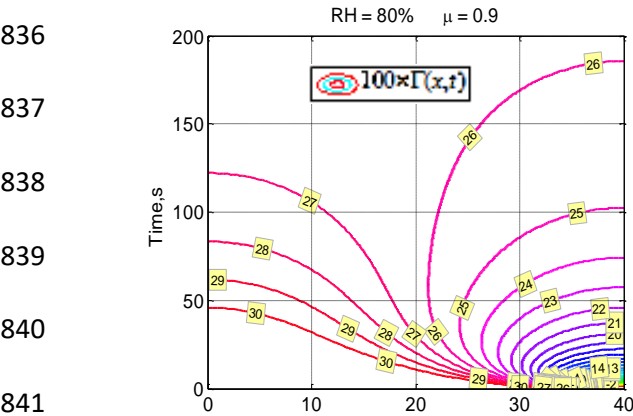
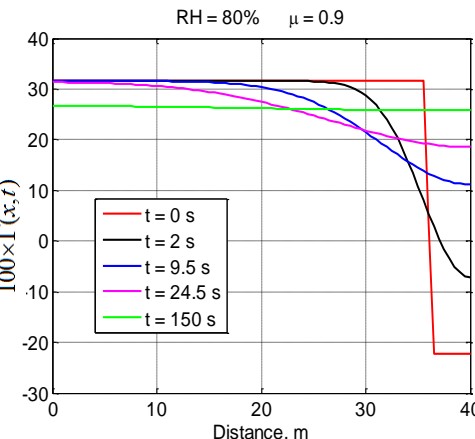

**Fig. 2.** Spatial-temporal variations of conservative function $100 \times \Gamma(x,t)$ for different cloud
fractions $\mu$ and initial $RH_2 = 80$ %.



















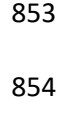
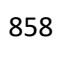

**Fig. 3.** Time evolution of DSD in the centers of the initially cloudy volume (left) and of the

initially dry air volume (right) at initially narrow DSD. The initial mixing parameters are

$RH_2 = 80\%$, $T = 10^{\circ}C$, $p = 828.8$ mb and $L = 40$ m.


















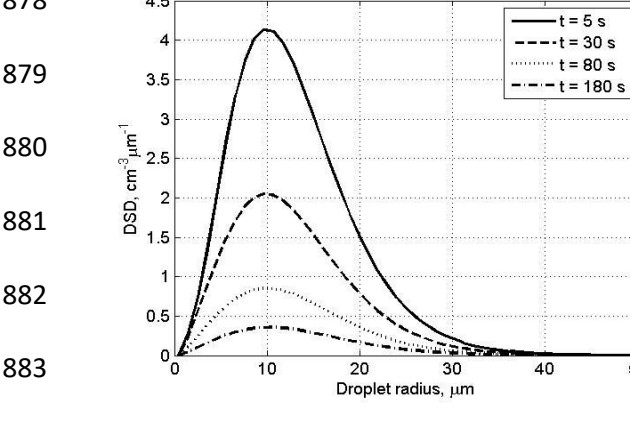
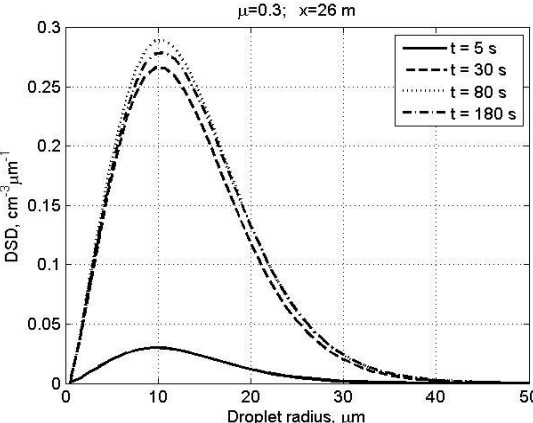







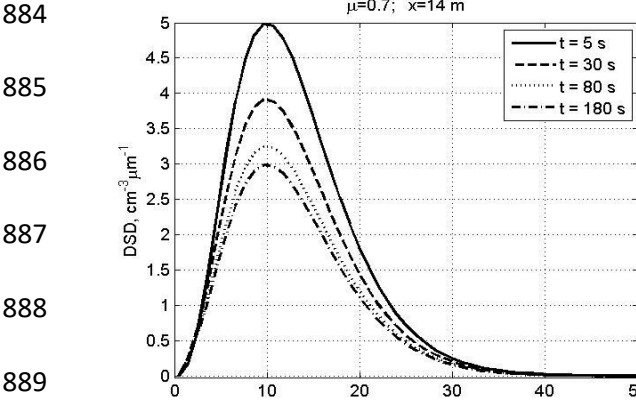
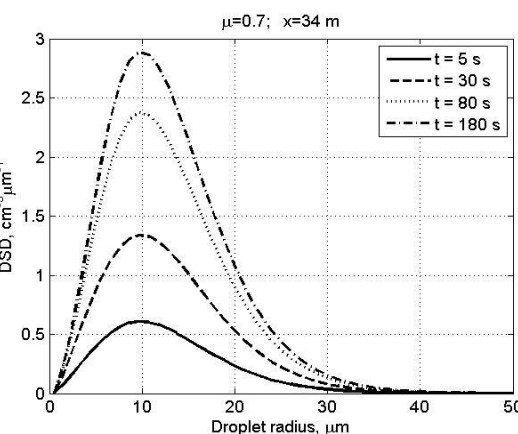



**Fig. 4.** The same as in Fig. 3, but for the initially wide DSD.


























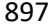

**Fig. 5.** Spatial dependences of droplet concentration, LWC and the mean volume radius

within the mixing volume at different time instances at narrow initial DSD. The initial mixing

parameters are $RH_2 = 80\,\%$, $T = 10^{\circ}C$, $p = 828.8$ mb and $L = 40\,\text{m}$.


























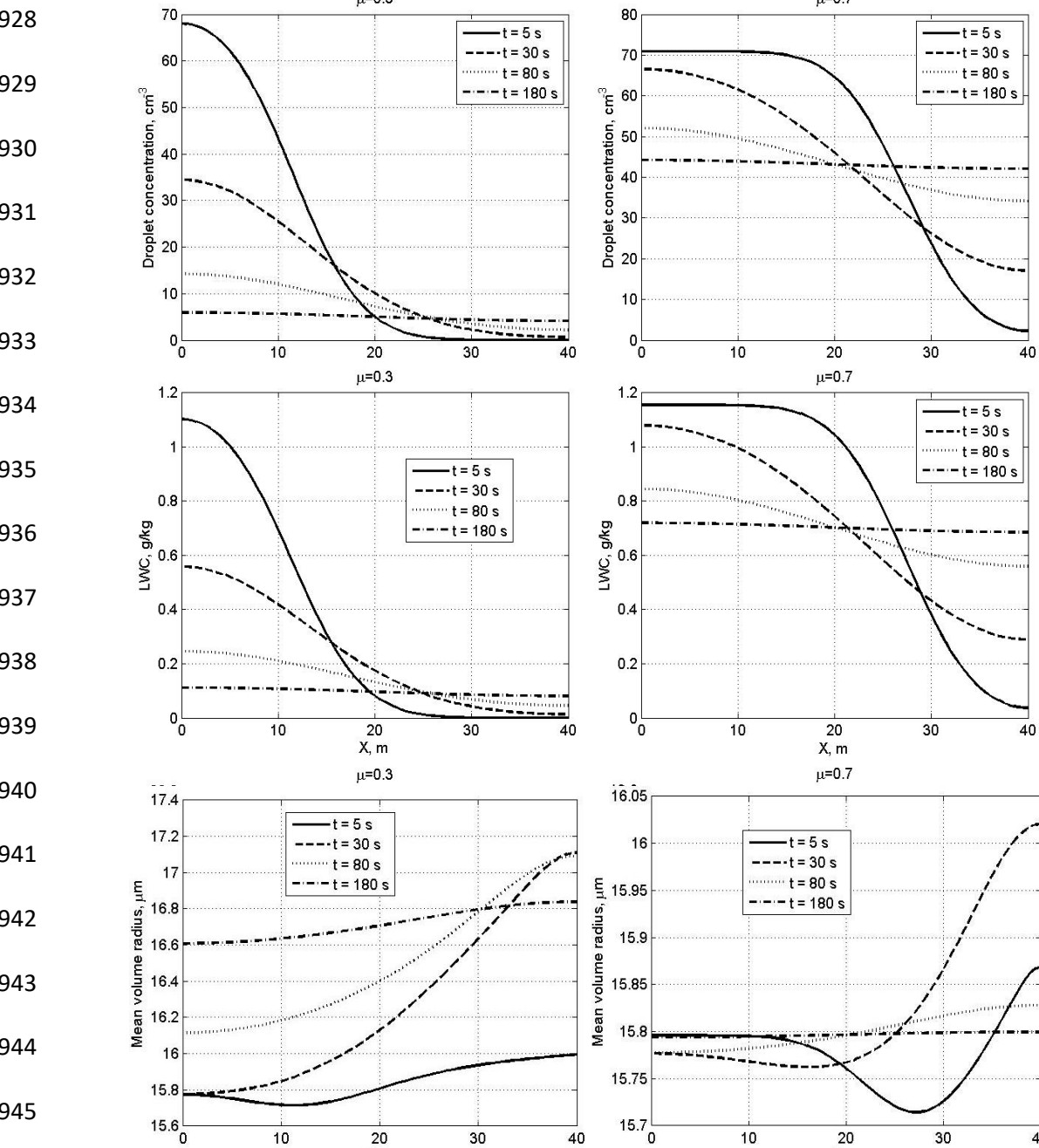

**Fig. 6.** The same as in Fig. 5, but for wide DSD











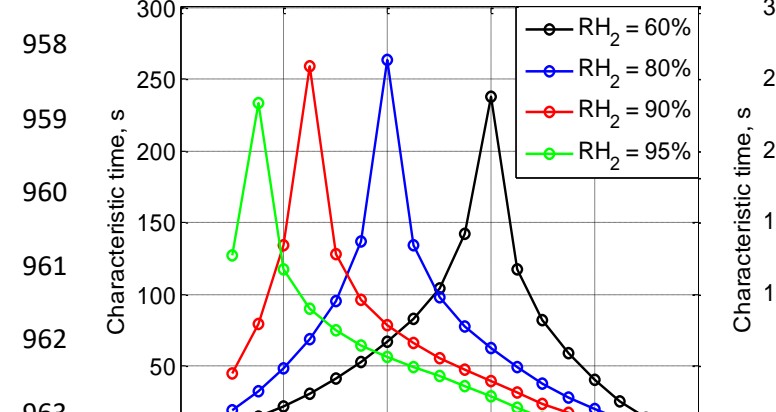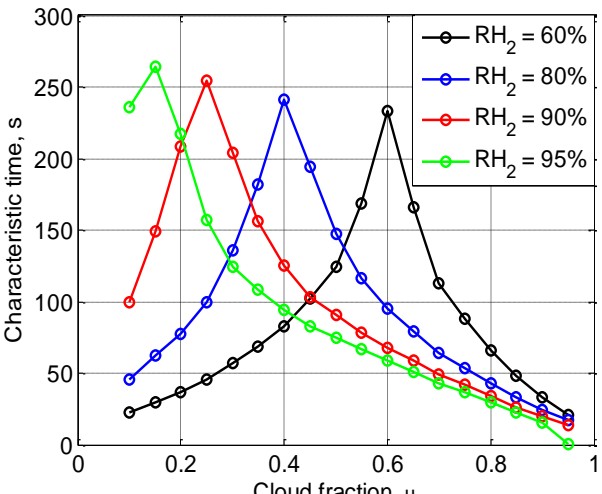

965

966

**Fig. 7.** Time required to reach the equilibrium state vs. the cloud fraction at different initial RH for the initially narrow DSD (left) and the initially wide DSD (right). Parameters of DSD are given in Tab. 1. The initial mixing parameters are $T = 10\,^{\circ}C$, $p = 828.8$ mb and $L = 40$ m.












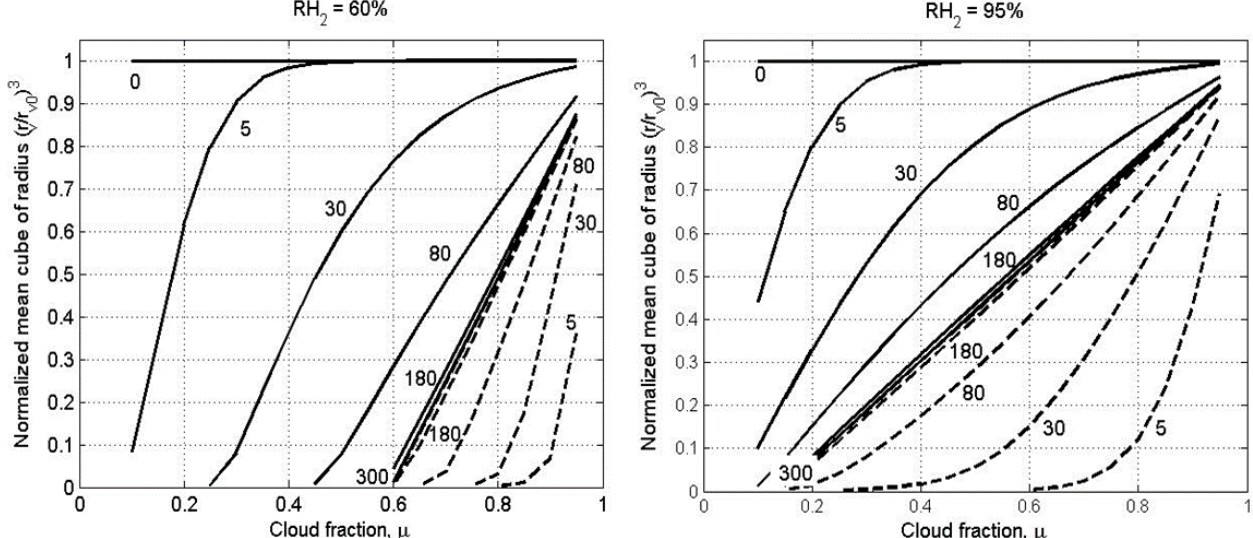



**Fig. 8.** Dependences of normalized cube of the mean volume radius on the cloud fraction
at different time instances for $x = 0$ (solid lines) corresponding to the initially cloud volume,
and $x = L$ (dash line) corresponding to the initially dry volume. The time instances in seconds
are marked by numbers. The figure is plotted for the narrow initial DSD for two values of
$RH_2$: 60% (left panel) and 95% (right panel). Parameters of DSD are given in Tab. 1. The
initial mixing parameters are $T = 10^{\circ}C$, $p = 828.8$ mb and $L = 40$ m. Calculations performed
within the range of $0.1 < \mu < 0.95$.








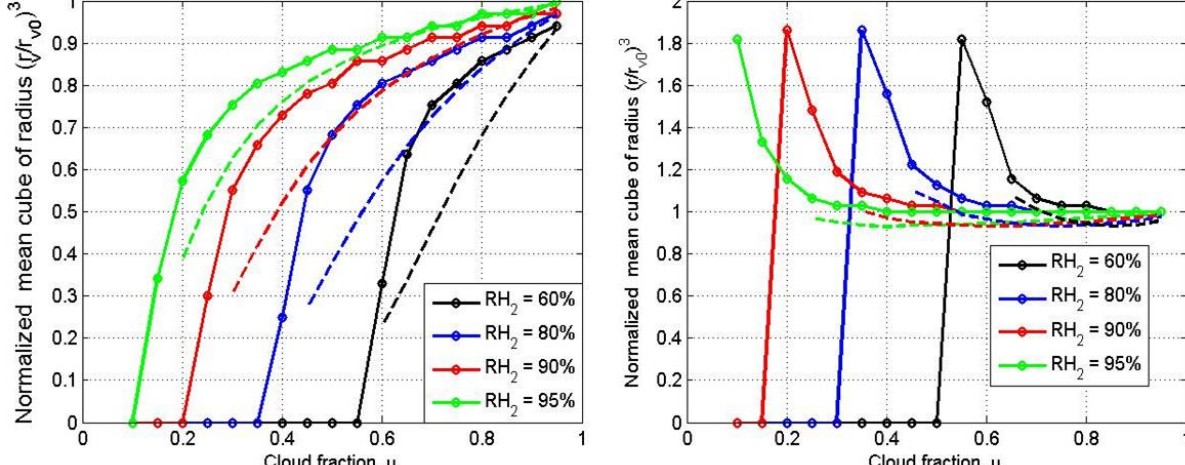




**Fig. 9.** Mixing diagrams. Normalized cube of the mean volume radius vs. the cloud fraction for initial narrow DSD (left) and initial wide DSD (right). The dependencies correspond to the equilibrium state. Parameters of initial DSD are presented in Tab. 1. Solid and dashed lines show the mixing diagrams for inhomogeneous and homogeneous mixing, respectively. The initial mixing parameters are $T = 10^{o}C$, $p = 828.8$ mb and $L = 40$ m.
























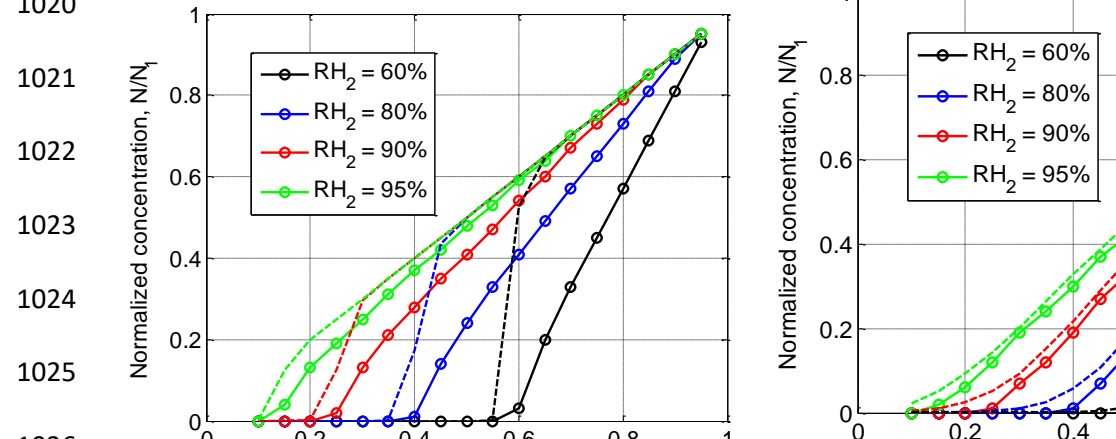

**Fig. 10.** Final normalized droplet concentration vs. cloud fraction for initially narrow DSD
(left) and initially wide DSD (right). Parameters of initial DSD are shown in Tab. 1. Dashed
line shows the results of equivalent homogeneous mixing. The initial mixing parameters are
$T = 10^{\circ}C$, $p = 828.8$ mb and $L = 40$ m.





















**Fig. 11.** Dependencies of normalized cube of the effective radius on normalized droplet
concentration for different initial relative humidity values. Left panel: narrow initial DSD.
Right panel: wide initial DSD. The initial mixing parameters are $T = 10^{\circ}C$, $p = 828.8$ mb
and $L = 40\,\text{m}$.

