# Peer review of "Theoretical analysis of mixing in liquid clouds. Part IV: DSD evolution and mixing diagrams"

_Atmospheric Chemistry and Physics, 2017_

## Referee Comment (RC1) · Anonymous Referee #1 · 12 Aug 2017

**Review of ACP-2017-508**

In this manuscript the authors made theoretical assessment of how mixing of cloudy air with unsaturated clear air affects the evolution of cloud microphysical parameters in the mixed volume, using a one dimensional diffusion-evaporation model. Entrainment and mixing in clouds have been known to be an important but not well understood problem in cloud physics for several decades even though their effects on cloud microphysics critically affect further cloud development. In that sense, this type of study is highly needed for advancement of our understanding of the problem. The authors calculated droplet spectral evolution in a one dimensional horizontal column of 40 m length during the turbulent mixing of cloudy air and unsaturated clear air of different proportions. Mixing starts at the interface between cloudy and clear air and proceeds gradually throughout the whole length by turbulent diffusion. Cloud droplet spectrum in the mixed volume changes due to droplet evaporation until saturation is restored. All these processes are expected to occur during the entrainment and mixing and with the model they employed, the authors seem to have calculated these processes well. But the critical question is if indeed they occur in real clouds under the conditions provided in this manuscript. If not, it would be inappropriate to give so much credit to the arguments the authors made in this manuscript. Nevertheless, I think that this manuscript is worth the publication if the authors clearly specify the limitation and applicability of their results. English is definitely not up to the standard of ACP publication and therefore requires great improvement. Some specific comments are followed.

**Major comments**

According to the model description, turbulent diffusion and evaporation of the droplets in the mixed volume occur simultaneously. The authors call this process inhomogeneous mixing because the degree of mixing is not uniform throughout the whole volume during the mixing. I am not sure if inhomogeneous mixing is the right term for this process. But I will stick to this definition for this review. In a one dimensional column, mixing may proceed only this way no matter what Damkohler number (Da) is. It seems that the model is capable of simulating homogeneous mixing as a special case for very small Da. How about the case of extremely inhomogeneous mixing, which is likely to occur when Da is very large? In a 3-D space overturning of turbulent eddies during the mixing of cloudy air with clear air may create some

portion of cloud volume remaining unaffected and some other portion of cloud volume losing all droplets due to complete evaporation. Is the model capable of simulating something similar to this 3-D reality when Da is very large? Or setting a different Da value just changes the speed of mixing and evaporation that occur in an invariably simultaneous fashion in the mixed volume? If so, this model does have intrinsic limitation. Another important point is that it takes nearly 5 min to arrive at the equilibrium state in the model simulations. In reality the mixed volume of 40 m length would not remain as an adiabatic entity for that long time as is pointed out by the authors themselves in the Discussion and Conclusion section. Therefore, I am hesitant to give too much credit for the arguments based on the results obtained at the equilibrium state.

Narrow and wide DSDs are used as input to the model. It is shown that mixing and evaporation actually result in increase of  $r_e$  in the mixed volume when DSD is wide. However, assuming a wide DSD does not reflect the reality. In the argument of entrainment and mixing, we start with an adiabatic cloud parcel and see how entrain and mixing of clear air would affect cloud microphysics in this parcel. Importantly, droplet distributions in adiabatic parcels are intrinsically narrow. The wide DSD the authors used is therefore unrealistic. If a wide DSD is observed in a cloud parcel, it would indicate that this parcel has already been through severe entrainment and mixing and/or coalescence process but certainly not a parcel that will start entrainment and mixing just now.

RH of 60% is the lowest in the model calculation of this manuscript. What about the mixing of air with RH of  $\sim$ 20%, which is a proper RH value for the air above the stratocumulus cloud top in the subtropics? Mixing will be more likely to be extremely inhomogeneous when this very dry air is entrained. Can this be simulated in the model?

The authors claim that mixing diagrams are not capable of distinguishing between mixing types. Moreover, results are not distinguishably different for different RH values of entrained clear air (Fig. 11). However, distinction between inhomogeneous and homogeneous mixing seems so obvious in Fig. 9a. It is not so in Fig. 9b as the authors claim but here unrealistically wide DSD is used for the calculation and therefore giving too much credit is unjustifiable. Even in the traditional mixing diagram that uses normalized concentration (N/N1) as in Fig. 11, distinction should be obvious between inhomogeneous and homogeneous mixing and also

among different RH values for homogeneous mixing. The authors do not put the lines for homogeneous mixing in Fig. 11. If they do, I expect that the results would be distinctively different from those for inhomogeneous mixing. Their argument is based only on the results for inhomogeneous mixing. However, RH dependence of inhomogeneous mixing is not known to be significant anyway and has never been discussed in mixing diagram analysis (e.g., Burnet and Brenguier, 2007). Another thing to note is that in the traditional mixing diagram, y-axis is the cube of normalized mean volume radius (representing mean volume of the droplets), not the cube of normalized effective radius as is used in this manuscript. With the obvious relationship of L = NV, where L is the liquid water content, N is the number concentration and V is the mean volume of the droplets,  $L/L_a$  (i.e., normalized L, where  $L_a$  is the adiabatic value of L) = const lines can be drawn with rectangular hyperbolic lines in the mixing diagram, making the diagram somewhat like a 3 D field of N, V and L (Burnet and Brenguier, 2007). So I recommend the authors to use mean volume radius instead of effective radius.

**Minor comments**

L28: It is not right to say that droplet concentration remains unchanged when mixing is homogeneous. It does reduce because of simple dilution of cloud volume by clear air. The total number of droplets in the whole mixing volume remains unchanged but not the concentration, which is the number of droplets in a unit volume.

In several Figure captions, it is stated that p = 8288 mb. Shouldn't it be 828.8 mb?

Figures 9, 10 and 11: How are  $r_e/r_{e0}$  values obtained? Are they averages for the whole mixed volume at the time when equilibrium is achieved? How about N/N1? Explain clearly.

Inappropriate English expressions are found in many places in this manuscript. They need to be corrected.

---

## Referee Comment (RC2) · B. Kumar (Referee) · 24 Aug 2017

**Theoretical analysis of mixing in liquid clouds. Part IV: DSD evolution and mixing diagrams by M. Pinsky and A. Khain**

This study present analysis of mixing in cloudy and clear air. Evolution of DSD has been analyzed using poly-disperse initial DSD and varying cloud fraction  $\mu$ . A diffusion-evaporation model was considered for the analysis. The main findings are

- Mixing diagram has multi-parameter characteristics.
- In dry volume, mixing leads to a rapid increase in RH
- DSD shape changes based on initial DSD chosen.
- The critical cloud fraction  $\mu_{cr}$  with respect to total droplet evaporation are same for any mixing type.
- Mixing diagrams for homogeneous and in-homogeneous mixing for poly-disperse DSD do not differ much.

It was concluded that classical concept of mixing diagram is too crude to distinguish the mixing type in observation data.

General comment: Overall, the manuscript is worth to publish after explaining questions below.

**Major comments:**

- 1. Explain the reason for considering droplet concentration by averaging along X-axis only why not in whole domain? Also, why vertical velocity was neglected? Since, the analysis is done based on these assumptions, it is inappropriate to make strong general statement about mixing diagrams.
- 2. In this analysis, collision and coalescence was not considered which also contribute in broadening of DSD. Authors should make comments on this issue.
- 3. The result shows that in dry volume large, droplets do not change their size significantly. This is not the case in general because during mixing, droplet size starts decreasing as soon as they enter in dry volume. Authors should provide the reason for it.
- 4. Traditional mixing diagrams should be plotted for normalized values of cube radii vs. number concentration and then compare with mixing diagrams proposed in this study.

**Minor comments:**

- 1. All figure labels, legends should be bigger size to be visible enough.
- 2. Some references related to recent numerical simulation of entrainment and mixing should be added.

---

## Author Comment (AC1) · 8 Sep 2017

Response to reviewer 1 We are grateful to Reviewer for his valuable comments and remarks. ─────────────────────────────────────

Review of ACP-2017-508

In this manuscript the authors made theoretical assessment of how mixing of cloudy air with unsaturated clear air affects the evolution of cloud microphysical parameters in the mixed volume, using a one dimensional diffusion-evaporation model. Entrainment and mixing in clouds have been known to be an important but not well understood

problem in cloud physics for several decades even though their effects on cloud microphysics critically affect further cloud development. In that sense, this type of study is highly needed for advancement of our understanding of the problem. The authors calculated droplet spectral evolution in a one dimensional horizontal column of 40 m length during the turbulent mixing of cloudy air and unsaturated clear air of different proportions. Mixing starts at the interface between cloudy and clear air and proceeds gradually throughout the whole length by turbulent diffusion. Cloud droplet spectrum in the mixed volume changes due to droplet evaporation until saturation is restored. All these processes are expected to occur during the entrainment and mixing and with the model they employed, the authors seem to have calculated these processes well. But the critical question is if indeed they occur in real clouds under the conditions provided in this manuscript. If not, it would be inappropriate to give so much credit to the arguments the authors made in this manuscript. Nevertheless, I think that this manuscript is worth the publication if the authors clearly specify the limitation and applicability of their results. English is definitely not up to the standard of ACP publication and therefore requires great improvement. Some specific comments are followed.

Major comments

© According to the model description, turbulent diffusion and evaporation of the droplets in the mixed volume occur simultaneously. The authors call this process inhomogeneous mixing because the degree of mixing is not uniform throughout the whole volume during the mixing. I am not sure if inhomogeneous mixing is the right term for this process. ® we state that mixing is inhomogeneous in a mixing volume because different droplets in the volume experience different subsaturations and evaporate with different rates. We believe that this definition is the logical and widely accepted.

© But I will stick to this definition for this review. In a one dimensional column, mixing may proceed only this way no matter what Damkohler number (Da) is. It seems that the model is capable of simulating homogeneous mixing as a special case for very small

Da. How about the case of extremely inhomogeneous mixing, which is likely to occur when Da is very large?

® Yes, we agree that the diffusional-evaporation model can describe both homogeneous and inhomogeneous mixing. Effects of Damkohler number on mixing within the frame of diffusional –evaporation model is analyzed in detail in Parts 2 and especially, in Part 3 of the study (Pinsky et al., 2016a,b). In particular, In Part 3 several types of mixing (homogeneous, intermediate, inhomogeneous and extremely inhomogeneous) are distinguished. In the present study we use parameters of mixing volume, turbulent diffusion and droplet size distribution, which as we suppose, are typical of real clouds. These parameters correspond to the values of Da of several hundred.

© In a 3-D space overturning of turbulent eddies during the mixing of cloudy air with clear air may create some portion of cloud volume remaining unaffected and some other portion of cloud volume losing all droplets due to complete evaporation. Is the model capable of simulating something similar to this 3-D reality when Da is very large? Or setting a different Da value just changes the speed of mixing and evaporation that occur in an in variably simultaneous fashion in the mixed volume? If so, this model does have intrinsic limitation.

® We do not describe formation of separate turbulent filaments. In the study we describe averaged effects of turbulent mixing by modeling of turbulent diffusion, characterized by a typical value of turbulent diffusion coefficient. This is specially stressed in the paper. In principle, the model allows consideration of penetration and mixing of separate filaments by setting the specific initial conditions. However, we suppose that spatial-time distribution of cloudy and droplet free filaments is not well known and the investigation of the sensitivity of mixing to such distributions is out of scope of the present study.

© Another important point is that it takes nearly 5 min to arrive at the equilibrium state in the model simulations. In reality the mixed volume of 40 m length would not remain

as an adiabatic entity for that long time as is pointed out by the authors themselves in the Discussion and Conclusion section. Therefore, I am hesitant to give too much credit for the arguments based on the results obtained at the equilibrium state.

® We agree with the reviewer. Moreover, it is the goal of this and some other our papers to show that mixing in real clouds does not reach equilibrium state and that the scattering diagrams observed in situ are just snapshots of the transient mixing process. However, since the classic mixing diagrams are plotted namely for equilibrium states, we investigate transition to such equilibrium assuming that the mixing volume remains adiabatic (i.e. isolated) during the entire period of mixing. This point is stressed in the revised paper.

© Narrow and wide DSDs are used as input to the model. It is shown that mixing and evaporation actually result in increase of re in the mixed volume when DSD is wide. However, assuming a wide DSD does not reflect the reality. In the argument of entrainment and mixing, we start with an adiabatic cloud parcel and see how entrain and mixing of clear air would affect cloud microphysics in this parcel. Importantly, droplet distributions in adiabatic parcels are intrinsically narrow. The wide DSD the authors used is therefore unrealistic. If a wide DSD is observed in a cloud parcel, it would indicate that this parcel has already been through severe entrainment and mixing and/or coalescence process but certainly not a parcel that will start entrainment and mixing just now.

® Indeed, pure diffusion growth leads to very narrow DSD. However, several other microphysical processes lead to DSD broadening. Mechanisms of DSD broadening in ascending adiabatic volumes are considered in several studies (e.g., Khain et al. (2000), Pinky and Khain (2002), Segal et al. (2004), Prabha et al. (2011)). These studies show that the DSD broadening is caused by in-cloud nucleation of droplets within clouds as well as by collisions. It is shown that DSDs in adiabatic volumes can be wide and first raindrops or drizzle arise in non-diluted adiabatic cloud parcels

(Khain et al., 2013, Magaritz-Ronen et al., 2016). We use wide DSD in the form of gamma distribution with parameters typically used in different bulk-parameterization models to describe wide and narrow DSDs. We agree that mixing leads to additional DSD broadening. We also do not see any problem if the DSDs, which are used as initial DSDs in cloudy parcels were affected by mixing during their previous history. It does not affect our analysis. The main point is that in these parcels initially RH=100%. Corresponding comments are included into the revised text.

© RH of 60% is the lowest in the model calculation of this manuscript. What about the mixing of air with RH of∼20%, which is a proper RH value for the air above the stratocumulus cloud top in the subtropics? Mixing will be more likely to be extremely inhomogeneous when this very dry air is entrained. Can this be simulated in the model?

® The model can work at any initial RH in the dry volume. At the same time very low RH leads to total evaporation of droplets in the mixing volume. Cloud fraction should exceed 0.8 to get droplets in the equilibrium state at RH=20% (at LWC=1 g/kg). We believe that turbulence above the stratocumulus cloud top is very weak so, cloud fraction should be large. At the lateral edges of warm Cu a shell of humid air arises around cloud, so RH of the entrained air is higher than 20% (Gerber et al. 2008).

© The authors claim that mixing diagrams are not capable of distinguishing between mixing types. Moreover, results are not distinguishably different for different RH values of entrained clear air (Fig. 11). However, distinction between inhomogeneous and homogeneous mixing seems so obvious in Fig. 9a. It is not so in Fig. 9b as the authors claim but here unrealistically wide DSD is used for the calculation and therefore giving too much credit is unjustifiable. Even in the traditional mixing diagram that uses normalized concentration (N/N1) as in Fig. 11, distinction should be obvious between inhomogeneous and homogeneous mixing and also among different RH values for homogeneous mixing. The authors do not put the lines for homogeneous mixing in Fig. 11. If they do, I expect that the results would be distinctively different from those for

inhomogeneous mixing. Their argument is based only on the results for inhomogeneous mixing. However, RH dependence of inhomogeneous mixing is not known to be significant anyway and has never been discussed in mixing diagram analysis (e.g., Burnet and Brenguier, 2007).

® The dispersion of points in situ measured mixing scattering diagrams is large. In the revised paper we present as an example a scattering diagram taken from Burnet and Brenguier (2007) with overloaded curves in Fig.9a (narrow DSD). Solid lines correspond to inhomogeneous mixing, while dashed lines correspond to homogeneous mixing. One can see that the high scattering does not allow to separate the mixing types.

© Another thing to note is that in the traditional mixing diagram, y-axis is the cube of normalized mean volume radius (representing mean volume of the droplets), not the cube of normalized effective radius as is used in this manuscript. With the obvious relationship of L = NV, where L is the liquid water content, N is the number concentration and V is the mean volume of the droplets, L/La (i.e., normalized L, where La is the adiabatic value of L) = const lines can be drawn with rectangular hyperbolic lines in the mixing diagram, making the diagram somewhat like a 3 D field of N, V and L (Burnet and Brenguier, 2007). So I recommend the authors to use mean volume radius instead of effective radius.

® We agree with Reviewer that the mean volume radius could be used in the analysis instead of effective radius. Note, however, that since effective radius in a wide range of conditions is only by 10% larger than the mean volume radius, the utilization of effective radius does not affect significantly the results. Moreover, satellites measure specifically effective radius. Accordingly, some authors (e.g. Freud et al., 2011) use effective radius for plotting the mixing diagrams.

Minor comments

© L28: It is not right to say that droplet concentration remains unchanged when mixing

is homogeneous. It does reduce because of simple dilution of cloud volume by clear air. The total number of droplets in the whole mixing volume remains unchanged but not the concentration, which is the number of droplets in a unit volume.

® Yes, we agree. The corresponding sentence is corrected (line 28).

© In several Figure captions, it is stated that p = 8288 mb. Shouldn't it be 828.8 mb?

® Corrected

© Figures 9, 10 and 11: How are re/re0 values obtained? Are they averages for the whole mixed volume at the time when equilibrium is achieved? How about N/N1? Explain clearly.

® In these figures the mixing diagrams corresponding to the final equilibrium states are calculated. In the equilibrium state (about 300s) all quantities, including effective radius and droplet concentration become uniform throughout entire volume.

© Inappropriate English expressions are found in many places in this manuscript. They need to be corrected. ® We have improved English whenever possible.

Response to comments and remarks of Dr. Kumar ————————————————
——— We are grateful to Dr. Kumar for valuable comments and remarks.

Theoretical analysis of mixing in liquid clouds. Part IV: DSD evolution and mixing diagrams by M. Pinsky and A. Khain

This study present analysis of mixing in cloudy and clear air. Evolution of DSD has been analyzed using poly-disperse initial DSD and varying cloud fraction $\mu$. A diffusion-evaporation model was considered for the analysis. The main findings are • Mixing diagram has multi-parameter characteristics. • In dry volume, mixing leads to a rapid increase in RH • DSD shape changes based on initial DSD chosen. • The critical cloud fraction $\mu$cr with respect to total droplet evaporation are same for any mixing type.

• Mixing diagrams for homogeneous and in-homogeneous mixing for poly-disperse DSD do not differ much.

It was concluded that classical concept of mixing diagram is too crude to distinguish the mixing type in observation data. [R] Thank you for the clear summary of the study

General comment: Overall, the manuscript is worth to publish after explaining questions below.

Major comments:

1. Explain the reason for considering droplet concentration by averaging along X-axis only why not in whole domain? Also, why vertical velocity was neglected? Since, the analysis is done based on these assumptions, it is inappropriate to make strong general statement about mixing diagrams.

[R] The paper reconsiders the classical theory of mixing diagrams. In the classical theory two volumes (cloudy and droplet free) mix with each other within a given unmovable mixing volume (see review by Korolev et al., 2016). The 1D diffusion- evaporation model is used in analysis. In this model all variables change along horizontal x-axes (e.g. figs. 2-7). This model does not involve any spatial averaging. The vertical velocity is also neglected in 1D model. Mixing diagrams are plotted for times when all variables become uniform within the mixing volume, i.e when the equilibrium state is reached.

We plot the mixing diagram applying widely used simplifications, namely: no vertical motions and no collisions. These assumptions allow to reveal better the microphysical effects of turbulent mixing. It is widely assumed that the mixing type is determined by the Damkohler number that the ratio of mixing time and drop relaxation time. Vertical velocity and collision rate are not included into this criterion. We extend the classical theory in several important aspects concerning microphysical effects: a) we consider time dependent process of mixing and b) initial droplet size distributions are assumed polydisperse. These simplifications are clearly formulated in the paper. We agree

that averaged vertical velocity, as well as collisions, affect DSDs, but these changes in DSDs are not related to mixing, and are described by other microphysical equations.

2. In this analysis, collision and coalescence was not considered which also contribute in broadening of DSD. Authors should make comments on this issue. ® The corresponding comments are included in the discussion section. We agree that many microphysical processes lead to the DSD broadening. The new feature that we stress in the study is that any mixing leads to DSD broadening (in contrast to conclusions of classical theory, considering monodisperse DSDs.) The corresponding comment is added into the conclusion section.

3. The result shows that in dry volume large, droplets do not change their size significantly. This is not the case in general because during mixing, droplet size starts decreasing as soon as they enter in dry volume. Authors should provide the reason for it. ® According to equation of diffusion growth/evaporation the rate of droplet radii in sub-saturation conditions decreases is inverse proportional to droplet radius. It means that if, say, 2 um radius droplet decreases twice during a certain time instance, the radius of 20 um droplet will decrease by less than 0.1 um. It means that relative decrease in the sizes of large droplets is much lower than that of small ones. The initial dry volume can faster saturate that hiders further evaporation of larger droplets. Actually reduction in larger droplet sizes can be insignificant. That is why we wrote that the size of large droplets remains approximately unchanged.

4. Traditional mixing diagrams should be plotted for normalized values of cube radii vs. number concentration and then compare with mixing diagrams proposed in this study.

® Traditional (classical) mixing diagrams are plotted for monodisperse DSDs. In this case the cloud fraction is equivalent to the normalized concentration. In the present study we plotted scattering and mixing diagrams both as dependencies of normalized values of cube radii on cloud fraction (figs. 8-10) and on number concentration (Fig 11)

[Figure]

Minor comments: 1. All figure labels, legends should be bigger size to be visible enough. ® improved

2. Some references related to recent numerical simulation of entrainment and mixing should be added.

® References to recent studies (Bera et al., 2016a,b; Kumar et al., 2014; 2017; Khain et al, 2017, Yum et al., 2016) are added.

Please also note the supplement to this comment:
https://www.atmos-chem-phys-discuss.net/acp-2017-508/acp-2017-508-AC1-supplement.pdf

---

## Referee Report (RR1)

**Theoretical analysis of mixing in liquid clouds. Part IV: DSD evolution and mixing diagrams   by M. Pinsky and A. Khain**

This study present analysis of mixing in cloudy and clear air.  Evolution of DSD has been analyzed using poly-disperse initial DSD and varying cloud fraction µ.  A diffusion-evaporation model was considered for the analysis. The main findings are

**General comments:**

The Authors have clarified almost all the questions raised.

 They have improved the all figure captions and legends. Also, reference section has been improved.

Overall, paper is fine for publication in ACP.

---

## Referee Report (RR2)

**Review of ACP-2017-508_revised**

In this revised version of the manuscript the authors addressed most of the comments I made to the original manuscript. The main points are clearer and their modeling results reveal some important aspects of entrainment and mixing processes in clouds. Now I think that this manuscript is worth the publication in ACP. However, I still think that the authors should clearly state the limitation of their results not to mislead the readers. English is improved significantly but the many typos I noticed should be corrected. It may not be necessary to review the manuscript again after corrections but the editor should check if the authors address my comments before making the final decision. Some specific comments are followed.

**Major comments**

One of the main arguments the authors make is the inappropriateness of the mixing diagram as a tool to analyze entrainment and mixing problem in clouds. Their argument is based on the fact that the mixing diagrams that can be drawn when equilibrium is reached in their model calculation are different from what is expected from the 'classical' mixing diagram for a particular mixing type, specifically inhomogeneous mixing at equilibrium state. This is misleading. When we draw mixing diagram, we do not assume anything. As the authors themselves state clearly several times, mixing diagrams of in-situ observation data just give us a snapshot of cloud microphysical relationships. We may assume equilibrium state only when we interpret the results, saying, for example, that such data scatter resembles something that can be expected from the final equilibrium state of inhomogeneous mixing or something that can reveal homogeneous mixing at its final stage. Even though mixing diagrams give us only the snapshot of different stages of entrainment and mixing process, they can still reveal some important information on the nature of entrainment and mixing process. That is the basic stance when we interpret mixing diagrams. In their response to my comments on the original manuscript, the authors showed two figures from Burnet and Brenguier (2007) that might demonstrate the difficulty of interpreting mixing diagram. The authors did not show another figure from Burnet and Brenguier (2007) that can indeed demonstrate clear difference of data scatter from the two figures the authors showed in their response to my comments, because this figure indicated inhomogeneous mixing unlike the two figures that indicated homogeneous mixing.

The authors should state the limitation of their model more clearly since this is the main reason why their results are different from observation. In real clouds, entrainment and mixing do not proceed continuously until the equilibrium state is reached as was postulated in their model. Intermittency certainly exists in real clouds as demonstrated in many observational studies by abrupt changes of droplet number concentrations near cloud edge regions but this cannot be generated with their model. Mixing diagram of in-situ observation data is a snapshot of cloud microphysical relationships that contains all these effects at an instance. The 'classical' mixing type idea is just one way of interpretation of mixing diagram. What if relative mean volume diameters do not change despite a large variation of relative droplet concentrations in a mixing diagram? A reasonable interpretation would be the dominance of inhomogeneous mixing for this cloud. What if relative mean volume diameters and relative droplet concentrations show a strong positive correlation? A reasonable interpretation would be the dominance of homogeneous mixing instead of ambiguity between homogeneous and inhomogeneous mixing. For inhomogeneous mixing would not continue until the equilibrium state is reached in real clouds and therefore mixing diagram would not become so similar between homogeneous and inhomogeneous mixing as the authors suggested with their model results. What if the data scatter does not suggest any of the 'classical' mixing type idea? A reasonable interpretation would be that some other processes must have been dominant.

The authors discussed some of these aspects in the last two paragraphs of Discussion and conclusion but their stance is still that mixing diagram is at fault. It is not that "classical mixing diagrams are plotted namely for equilibrium states." Mixing diagram is not plotted for anything but in the interpretation of mixing diagram we may adopt the concept of inhomogeneous or homogeneous mixing at equilibrium state. The authors should first emphasize the limitation of their modeling results more clearly and then the cautions we may take when we interpret mixing diagrams of in-situ observation data.

**Minor comments**

DSD is a collective term. So the word "DSD maximum" seems awkward at Line 313 and at several other lines. More appropriate expression seems to be the mode diameter of DSD. Similarly what does "DSD values" mean? Collectively it would mean total droplet

concentration. Make it clear.

There are many typos. One example is "within in the initially dry volume" at Line 326. These should be corrected.

---

## Author Response (AR2)

8 November, 2017

Dear Prof. Garrett,

Please find attached the revised paper "Theoretical analysis of mixing in liquid clouds. Part IV: DSD evolution and mixing diagrams", authored by Mark Pinsky, and Alexander Khain. All Your comments and remarks, as well as comments and remarks of reviewers are carefully addressed. Line-by-line responses are attached as well as the MS with the changes highlighted.

In considerations, the effective radius was replaced by the mean volume radius. Accordingly, seven figures were replotted. Explanations given in the responses to reviewers are given now in the text of the revised paper.

Sincerely yours,

Alexander

Prof. Alexander Khain

 Department of Atmospheric Sciences,

The Hebrew University of Jerusalem

Israel

**Response to reviewer 1**
We are grateful to Reviewer for his valuable comments and remarks.

**Review of ACP-2017-508**

In this manuscript the authors made theoretical assessment of how mixing of cloudy air with unsaturated clear air affects the evolution of cloud microphysical parameters in the mixed volume, using a one dimensional diffusion-evaporation model. Entrainment and mixing in clouds have been known to be an important but not well understood problem in cloud physics for several decades even though their effects on cloud microphysics critically affect further cloud development. In that sense, this type of study is highly needed for advancement of our understanding of the problem. The authors calculated droplet spectral evolution in a one dimensional horizontal column of 40 m length during the turbulent mixing of cloudy air and unsaturated clear air of different proportions. Mixing starts at the interface between cloudy and clear air and proceeds gradually throughout the whole length by turbulent diffusion. Cloud droplet spectrum in the mixed volume changes due to droplet evaporation until saturation is restored. All these processes are expected to occur during the entrainment and mixing and with the model they employed, the authors seem to have calculated these processes well. But the critical question is if indeed they occur in real clouds under the conditions provided in this manuscript. If not, it would be inappropriate to give so much credit to the arguments the authors made in this manuscript. Nevertheless, I think that this manuscript is worth the publication if the authors clearly specify the limitation and applicability of their results. English is definitely not up to the standard of ACP publication and therefore requires great improvement. Some specific comments are followed.

**Major comments**

© According to the model description, turbulent diffusion and evaporation of the droplets in the mixed volume occur simultaneously. The authors call this process inhomogeneous mixing because the degree of mixing is not uniform throughout the whole volume during the mixing. I am not sure if inhomogeneous mixing is the right term for this process.

® We state that mixing is inhomogeneous in a mixing volume because different droplets in the volume experience different subsaturations and evaporate with different rates. We believe that this definition is the logical and widely accepted.

© But I will stick to this definition for this review. In a one dimensional column, mixing may proceed only this way no matter what Damkohler number (Da) is. It seems that the model is capable of simulating homogeneous mixing as a special case for very small Da. How about the case of extremely inhomogeneous mixing, which is likely to occur when Da is very large?

® Yes, we agree that the diffusional-evaporation model can describe both homogeneous and inhomogeneous mixing. Effects of Damkohler number on mixing within the frame of diffusional – evaporation model is analyzed in detail in Part 2 and especially, in Part 3 of the study (Pinsky et al., 2016a,b). In particular, In Part 3 several types of mixing (homogeneous, intermediate, inhomogeneous and extremely inhomogeneous) are distinguished. In the present study we use parameters of mixing volume, turbulent diffusion and droplet size distribution, which as we suppose, are typical of **real** clouds. These parameters correspond to the values of Da of several hundred. Corresponding comments are included into the revised paper.

© In a 3-D space overturning of turbulent eddies during the mixing of cloudy air with clear air may create some portion of cloud volume remaining unaffected and some other portion of cloud volume losing all droplets due to complete evaporation. Is the model capable of simulating something similar to this 3-D reality when Da is very large? Or setting a different Da value just changes the speed of mixing and evaporation that occur in an in variably simultaneous fashion in the mixed volume? If so, this model does have intrinsic limitation.

® We do not describe formation of separate turbulent filaments. In the study we describe averaged effects of turbulent mixing by modeling of turbulent diffusion, characterized by a typical value of turbulent diffusion coefficient. This is specially stressed in the paper. In principle, the model allows consideration of penetration and mixing of separate filaments by setting the specific initial conditions. However, we suppose that spatial-time distribution of cloudy and droplet free filaments is not well known and require special consideration. Such investigations are out of scope of the present study.

© Another important point is that it takes nearly 5 min to arrive at the equilibrium state in the model simulations. In reality the mixed volume of 40 m length would not remain as an adiabatic entity for that long time as is pointed out by the authors themselves in the Discussion and Conclusion section. Therefore, I am hesitant to give too much credit for the arguments based on the results obtained at the equilibrium state.

® We agree with Reviewer. Moreover, in this study and in some other our papers we show that mixing in real clouds does not reach equilibrium state and that the scattering diagrams observed in situ are just snapshots of the transient mixing process. However, since the classic mixing diagrams are plotted namely for equilibrium states, we investigate transition to such equilibrium assuming that the mixing volume remains adiabatic (i.e. isolated) during the entire period of mixing.
    This point is stressed in the revised paper (conclusion section).

© Narrow and wide DSDs are used as input to the model. It is shown that mixing and evaporation actually result in increase of $r_e$ in the mixed volume when DSD is wide. However, assuming a wide DSD does not reflect the reality. In the argument of entrainment and mixing, we start with an adiabatic cloud parcel and see how entrain and mixing of clear air would affect cloud microphysics in this parcel. Importantly, droplet distributions in adiabatic parcels are intrinsically narrow. The wide DSD the authors used is therefore unrealistic. If a wide DSD is observed in a cloud parcel, it would indicate that this parcel has already been through severe entrainment and mixing and/or coalescence process but certainly not a parcel that will start entrainment and mixing just now.

® Indeed, pure diffusion growth leads to very narrow DSD. However, several other microphysical processes lead to DSD broadening. Mechanisms of DSD broadening in ascending adiabatic volumes are considered in several studies (e.g., Khain et al (2000), Pinky and Khain (2002), Segal et al. (2004), Prabha et al. (2011). These studies show the DSD broadening is caused by in-cloud nucleation of droplets within clouds as well as by collisions. It is shown that DSDs in adiabatic volumes can be wide and first raindrops or drizzle arise namely in non-diluted adiabatic cloud parcels (Khain et al. 2013), Magaritz-Ronen et al. 2016). We use wide DSD in the form of gamma distribution with parameters typically used in different bulk-parameterization models. We agree that mixing leads to additional DSD broadening. We also do not see any problem if the DSDs, which are used as initial DSDs in cloudy parcels were affected by mixing during by their previous history. It does not affect our analysis.

The main point is that in these parcels initially RH=100%. Note that we have chosen quite wide DSD to reveal better the effect of DSD width.

Corresponding comments are included into the revised text.

© RH of 60% is the lowest in the model calculation of this manuscript. What about the mixing of air with RH of~20%, which is a proper RH value for the air above the stratocumulus cloud top in the subtropics? Mixing will be more likely to be extremely inhomogeneous when this very dry air is entrained. Can this be simulated in the model?

® The model can work at any initial RH in the dry volume. At the same time very low RH leads to total evaporation of droplets in the mixing volume. Cloud fraction should exceed 0.8 to get droplets in the equilibrium state at RH=20% (at LWC=1 g/kg). At the same time, we are interested in the cases, when droplets exist in the equilibrium state. At the lateral edges of warm Cu a shell of humid air arises around cloud, so RH of the entrained air is probably higher than 20%.

© The authors claim that mixing diagrams are not capable of distinguishing between mixing types. Moreover, results are not distinguishably different for different RH values of entrained clear air (Fig. 11). However, distinction between inhomogeneous and homogeneous mixing seems so obvious in Fig. 9a. It is not so in Fig. 9b as the authors claim but here unrealistically wide DSD is used for the calculation and therefore giving too much credit is unjustifiable. Even in the traditional mixing diagram that uses normalized concentration ($N/N_1$) as in Fig. 11, distinction should be obvious between inhomogeneous and homogeneous mixing and also among different RH values for homogeneous mixing. The authors do not put the lines for homogeneous mixing in Fig. 11. If they do, I expect that the results would be distinctively different from those for inhomogeneous mixing.

Their argument is based only on the results for inhomogeneous mixing. However, RH dependence of inhomogeneous mixing is not known to be significant anyway and has never been discussed in mixing diagram analysis (e.g., Burnet and Brenguier, 2007).

® The dispersion of points in situ measured mixing scattering diagrams is large. Below we present as an example a scattering diagram taken from Burnet and Brenguier (2007) with overloaded curves

in Fig.9a (narrow DSD). Solid lines correspond to inhomogeneous mixing, while dashed lines correspond to homogeneous mixing. One can see that the high scattering makes the separation between mixing types to be difficult problem.

[Figure]

© Another thing to note is that in the traditional mixing diagram, y-axis is the cube of normalized mean volume radius (representing mean volume of the droplets), not the cube of normalized effective radius as is used in this manuscript. With the obvious relationship of $L = NV$, where L is the liquid water content, N is the number concentration and V is the mean volume of the droplets, $L/L_a$ (i.e., normalized L, where $L_a$ is the adiabatic value of L) = const lines can be drawn with rectangular hyperbolic lines in the mixing diagram, making the diagram somewhat like a 3 D field of N, V and L (Burnet and Brenguier, 2007). So I recommend the authors to use mean volume radius instead of effective radius.

® The mean volume radius is used in the revised paper in the analysis instead of effective radius. Seven figures were replotted accordingly.

**Minor comments**

© L28: It is not right to say that droplet concentration remains unchanged when mixing is homogeneous. It does reduce because of simple dilution of cloud volume by clear air. The total number of droplets in the whole mixing volume remains unchanged but not the concentration, which is the number of droplets in a unit volume.

® Yes, we agree. The corresponding sentence is corrected.

© In several Figure captions, it is stated that p = 8288 mb. Shouldn't it be 828.8 mb?

® Corrected

© Figures 9, 10 and 11: How are re/re0 values obtained? Are they averages for the whole mixed volume at the time when equilibrium is achieved? How about N/N1? Explain clearly.

® In these figures the mixing diagrams corresponding to the final equilibrium states are calculated. In the equilibrium state (about 300s) all quantities, including mean volume radii (and effective radius) and droplet concentration become uniform along x-coordinate.

© Inappropriate English expressions are found in many places in this manuscript. They need to be corrected.
® We have improved English whenever possible.

**We are grateful to Dr. Kumar for valuable comments and remarks.**

**Theoretical analysis of mixing in liquid clouds. Part IV: DSD evolution**
**and mixing diagrams by M. Pinsky and A. Khain**

This study present analysis of mixing in cloudy and clear air. Evolution of DSD has been analyzed using poly-disperse initial DSD and varying cloud fraction $\mu$. A diffusion-evaporation model was considered for the analysis. The main findings are

• Mixing diagram has multi-parameter characteristics.
• In dry volume, mixing leads to a rapid increase in RH
• DSD shape changes based on initial DSD chosen.
• The critical cloud fraction $\mu_{cr}$ with respect to total droplet evaporation are same for any mixing type.
• Mixing diagrams for homogeneous and in-homogeneous mixing for poly-disperse DSD do not differ much.

It was concluded that classical concept of mixing diagram is too crude to distinguish the mixing type in observation data.
® Thank you for the clear summary of the study

**General comment**: Overall, the manuscript is worth to publish after explaining questions below.

**Major comments:**

1. Explain the reason for considering droplet concentration by averaging along X-axis only why not in whole domain? Also, why vertical velocity was neglected? Since, the analysis is done based on these assumptions, it is inappropriate to make strong general statement about mixing diagrams.

® The paper reconsiders the classical theory of mixing diagrams. In the classical theory two volumes (cloudy and droplet free) mix with each other within a given unmovable mixing volume (see review by Korolev et al., 2016). We *do not perform* averaging along X-direction. Instead, we calculate time dependencies of microphysical parameters along X-direction (e.g. figs. 2-7). Mixing diagrams are plotted for times when all variables become uniform within the mixing volume, i.e. when the equilibrium state is reached.
We plot the mixing diagram using the same simplifications used in the widely accepted (classical) mixing diagrams, namely: no vertical motions and no collisions are allowed. These assumptions allow to reveal better the microphysical effects of turbulent mixing. It is widely assumed that the mixing type is determined by the Damkohler number that depends only on drop relaxation time and mixing time. No averaged vertical velocity and no collision rate are included into this criterion.

Aiming at reconsidering the classical theory, we use the same simplifications as regards to dynamical processes (i.e. mixing is assumed to be turbulent, and averaged vertical velocity is equal to zero). We extend the theory, however, in several important aspects concerning

microphysical effects: a) we consider time dependent process of mixing and b) initial droplet size distributions are assumed polydisperse.

These simplifications are clearly formulated in the paper and corresponding discussion is included to section 4 of the revised paper.

We agree that averaged vertical velocity, as well as collisions, affect DSDs, but these changes in DSDs are not related directly to mixing, and are described by other microphysical equations.

2.    In this analysis, collision and coalescence was not considered which also contribute in broadening of DSD. Authors should make comments on this issue.

®. We agree that many microphysical processes lead to the DSD broadening. The new feature that we stress in the study is that any mixing leads to DSD broadening (in contrast to conclusions of classical theory, considering monodisperse DSDs.) The corresponding comment is added into the conclusion section.

3.    The result shows that in dry volume large droplets do not change their size significantly. This is not the case in general because during mixing, droplet size starts decreasing as soon as they enter in dry volume. Authors should provide the reason for it.

® According to equation of diffusion growth/evaporation, in of sub-saturation the rate of droplet radii decreases is inverse proportional to droplet radius. It means that if, say, 2 $\mu m$ radius droplet decreases twice during a certain time instance, the radius of 20 $\mu m$ droplet will decrease by less than 0.1 $\mu m$. It means that the relative decrease in the sizes of large droplets is much lower than that of small ones. That is why, we wrote that the size of large droplets remains approximately unchanged.

Corresponding comments are included into the revised paper (see description of the DSD evolution).

4.    Traditional mixing diagrams should be plotted for normalized values of cube radii vs. number concentration and then compare with mixing diagrams proposed in this study.

® All figures (seven figures) containing effective radius are replotted. The effective radius is replaced by the mean volume radius.

**Minor comments:**
1.    All figure labels, legends should be bigger size to be visible enough.
® Figures 5 and 6 are replotted.

2.    Some references related to recent numerical simulation of entrainment and mixing should be added.

® References to recent studies (Andrejczuk et al. 2006, 2009; Bera et al, 2016a,b; Kumar et al, 2014; 2017; Khain et al, 2017, Yum et al, 2016;) are included.

---

## Author Response (AR3)

We are grateful to Reviewer for the valuable comments and remarks.

**Review of ACP-2017-508_revised**

In this revised version of the manuscript the authors addressed most of the comments I made to the original manuscript. The main points are clearer and their modeling results reveal some important aspects of entrainment and mixing processes in clouds. Now I think that this manuscript is worth the publication in ACP. However, I still think that the authors should clearly state the limitation of their results not to mislead the readers. English is improved significantly but the many typos I noticed should be corrected. It may not be necessary to review the manuscript again after corrections but the editor should check if the authors address my comments before making the final decision. Some specific comments are followed.

**Major comments**

® General comment. To avoid the misinterpretation, we would like to make comment about the terminology used. We prefer to refer the diagrams plotted using observed data to as scattering diagrams in order to distinguish them from the theoretical mixing diagrams plotted for the hypothetical final equilibrium state.

© One of the main arguments the authors make is the inappropriateness of the mixing diagram as a tool to analyze entrainment and mixing problem in clouds. Their argument is based on the fact that the mixing diagrams that can be drawn when equilibrium is reached in their model calculation are different from what is expected from the 'classical' mixing diagram for a particular mixing type, specifically inhomogeneous mixing at equilibrium state. This is misleading.

© In the study we showed that
a)    The critical value of cloud fraction does not depend on the mixing type (eq. 11). The mixing diagrams should obey eq. (11) and should be plotted in agreement with eq. (11). In particular, the diagram for inhomogeneous mixing (i.e. horizontal straight line of Dv^3/Dva^3=1) should rapidly drop to zero at $\mu = \mu_{cr}$

b) The shape of mixing diagrams reached at the final equilibrium state depends on the initial shape of DSD. Accordingly, our mixing diagrams calculated for the final equilibrium state differ from the classical mixing diagrams calculated for monodisperse DSD.

We see nothing misleading in these results.

© When we draw mixing diagram, we do not assume anything.

® Yes, we agree that the *scattering* diagrams are plotted without assuming anything.

© As the authors themselves state clearly several times, mixing diagrams of in-situ observation data just give us a snapshot of cloud microphysical relationships. We may assume equilibrium state only when we interpret the results, saying, for example, that such data scatter resembles something that can be expected from the final equilibrium state of inhomogeneous mixing or something that can reveal homogeneous mixing at its final stage.

® Unfortunately, we cannot agree with the reviewer. As it is shown in the study, the scattering diagrams can dramatically differ from the final mixing diagram. So, it is quite difficult to assume that "such data scatter resembles something that can be expected from the final equilibrium state".

© Even though mixing diagrams give us only the snapshot of different stages of entrainment and mixing process, they can still reveal some important information on the nature of entrainment and mixing process. That is the basic stance when we interpret mixing diagrams. In their response to my comments on the original manuscript, the authors showed two figures from Burnet and Brenguier (2007) that might demonstrate the difficulty of interpreting mixing diagram. The authors did not show another figure from Burnet and Brenguier (2007) that can indeed demonstrate clear difference of data scatter from the two figures the authors showed in their response to my comments, because this figure indicated inhomogeneous mixing unlike the two figures that indicated homogeneous mixing.

® In our opinion based on the analysis of Damkohler number values and turbulent scales responsible for mixing in real clouds, mixing in real clouds is always inhomogeneous (Pinsky et al, 2016a; Khain et al, 2017).

As regards to scattering diagrams shown in Fig .8 from Burnet and Brenguier (2007), we agree that the shapes of scattering diagrams are different. At the same time, we cannot say that the panel (a) indicates inhomogeneous mixing, while two other panels indicate homogeneous mixing. For instance, scattering diagram in panel (a) can indicate that most of points correspond to slightly diluted cloudy volumes, or/and high humidity of entrained air, or/and significant width of DSDs.

The points located at larger distances from the horizontal straight line of $D_v^3/D_{va}^3=1$ in panels (b) and (c) correspond to air volumes with decreased droplet concentrations. These points likely indicate that the air volumes just entrained the cloud, as it is discussed in detail by Khain et al. (2017). Then these volumes mix inhomogeneously with surrounding cloud volumes. So, the large distance from the horizontal line $D_v^3/D_{va}^3=1$ does not indicate that the mixing is homogeneous.

At the same time, the scattering diagrams characterize the intensity of the process of entrainment and mixing, contain the information about DSD width and other DSD properties. We refer the study by Khain et al. (2017) for more detail. This paper is attached to our response. The corresponding comments concerning usefulness of scattering diagrams are added into the Conclusion section.

© The authors should state the limitation of their model more clearly since this is the main reason why their results are different from observation. In real clouds, entrainment and mixing do not proceed continuously until the equilibrium state is reached as was postulated in their model. Intermittency certainly exists in real clouds as demonstrated in many observational studies by abrupt changes of droplet number concentrations near cloud edge regions but this cannot be generated with their model. Mixing diagram of in-situ observation data is a snapshot of cloud microphysical relationships that contains all these effects at an instance.

® This study considers mixing in closed air volumes. The assumption automatically leads to establishing of final equilibrium state. We clearly stress in the paper that the equilibrium state cannot be reached in real clouds. This is one of the reasons of our statement that mixing diagrams hardly can be used for analysis of mixing type in clouds.

We agree that the intermittency of mixing is not described by the model, which is another reason why our results are different from observation.

The corresponding comment is included into the revised paper (Conclusion section).

© The 'classical' mixing type idea is just one way of interpretation of mixing diagram. What if relative mean volume diameters do not change despite a large variation of relative droplet concentrations in a mixing diagram? A reasonable interpretation would be the dominance of inhomogeneous mixing for this cloud. What if relative mean volume diameters and relative droplet concentrations show a strong positive correlation? A reasonable interpretation would be the dominance of homogeneous mixing instead of ambiguity between homogeneous and inhomogeneous mixing. For inhomogeneous mixing would not continue until the equilibrium state is reached in real clouds and therefore mixing diagram would not become so similar between homogeneous and inhomogeneous mixing as the authors suggested with their model results. What if the data scatter does not suggest any of the 'classical' mixing type idea? A reasonable interpretation would be that some other processes must have been dominant.

® This comment of Reviewer is largely of methodological nature. When one or another processes is not understood well, it is reasonable to use the simplest arguments for interpretation of observed data. However, as soon as observational and theoretical data allow deeper understanding of the process, the interpretations should be changed and improved.

© The authors discussed some of these aspects in the last two paragraphs of Discussion and conclusion but their stance is still that mixing diagram is at fault. It is not that "classical mixing diagrams are plotted namely for equilibrium states." Mixing diagram is not plotted for anything but in the interpretation of mixing diagram we may adopt the concept of inhomogeneous or homogeneous mixing at equilibrium state. The authors should first emphasize the limitation of their modeling results more clearly and then the cautions we may take when we interpret mixing diagrams of in-situ observation data.

® It seems that Reviewer means scattering diagrams in his comment. As we mentioned above, we refer the diagrams plotted using observed data to as scattering diagrams in order to distinguish them from the theoretical mixing diagrams plotted for the hypothetical final equilibrium state. In our conclusion we mean the limitations of the mixing diagrams plotted in the hypothetical final equilibrium state.

**Minor comments**

© DSD is a collective term. So the word "DSD maximum" seems awkward at Line 313 and at several other lines. More appropriate expression seems to be the mode diameter of DSD. Similarly, what does "DSD values" mean? Collectively it would mean total droplet concentration. Make it clear.

® Corrected

© There are many typos. One example is "within in the initially dry volume" at Line 326. These should be corrected.

® Corrected

[revised manuscript text omitted]